# Mosaic chromosomal alterations in peripheral blood leukocytes of children in sub-Saharan Africa

Weiyin Zhou[1,2], Anja Fischer [3], Martin D. Ogwang[4], Wen Luo[1,2], Patrick Kerchan[5], Steven J. Reynolds[6], Constance N. Tenge[7], Pamela A. Were[8], Robert T. Kuremu[7], Walter N. Wekesa[7], Nestory Masalu[9], Esther Kawira[10], Tobias Kinyera[4,11], Isaac Otim[4,11], Ismail D. Legason [5,11], Hadijah Nabalende[4,11], Leona W. Ayers[12], Kishor Bhatia[1], James J. Goedert[1], Mateus H. Gouveia [13], Nathan Cole[1,2], Belynda Hicks [1,2], Kristine Jones[1,2], Michael Hummel[14,15], Mathias Schlesner [16], George Chagaluka[17], Nora Mutalima[18,19], Eric Borgstein[17], George N. Liomba[17], Steve Kamiza[17], Nyengo Mkandawire[17], Collins Mitambo[20], Elizabeth M. Molyneux[17], Robert Newton [18], Selina Glaser[3], Helene Kretzmer[21], Michelle Manning[1,2], Amy Hutchinson[1,2], Ann W. Hsing[22], Yao Tettey[23], Andrew A. Adjei[23], Stephen J. Chanock [1,2], Reiner Siebert[3], Meredith Yeager [1,2], Ludmila Prokunina-Olsson [1], Mitchell J. Machiela [1] & Sam M. Mbulaiteye [1] ✉

In high-income countries, mosaic chromosomal alterations in peripheral blood leukocytes are associated with an elevated risk of adverse health outcomes, including hematologic malignancies. We investigate mosaic chromosomal alterations in sub-Saharan Africa among 931 children with Burkitt lymphoma, an aggressive lymphoma commonly characterized by immunoglobulin-*MYC* chromosomal rearrangements, 3822 Burkitt lymphoma-free children, and 674 cancer-free men from Ghana. We find autosomal and X chromosome mosaic chromosomal alterations in 3.4% and 1.7% of Burkitt lymphoma-free children, and 8.4% and 3.7% of children with Burkitt lymphoma (*P*-values = $5.7 \times 10^{-11}$ and $3.74 \times 10^{-2}$, respectively). Autosomal mosaic chromosomal alterations are detected in 14.0% of Ghanaian men and increase with age. Mosaic chromosomal alterations in Burkitt lymphoma cases include gains on chromosomes 1q and 8, the latter spanning *MYC*, while mosaic chromosomal alterations in Burkitt lymphoma-free children include copy-neutral loss of heterozygosity on chromosomes 10, 14, and 16. Our results highlight mosaic chromosomal alterations in sub-Saharan African populations as a promising area of research.

Mosaic chromosomal alterations (mCAs) are a type of clonal hematopoiesis detected in blood and characterized by large, structural copy number or copy-neutral loss of heterozygosity (CNLOH) events detected in a subset of cells[1]. In high-income countries, mCAs are detected in 0.2–1.0% of children undergoing clinical genetic testing for developmental disorders[2], but population-based investigations indicate a sharp increase in mCAs frequency in adults, from ~2% in individuals aged 40–60 years to 26.5% in those aged >80 years[3]. Little is

known about mCAs prevalence or distribution in normal peripheral blood leukocytes of individuals living in sub-Saharan Africa (SSA). Children in SSA suffer an elevated risk of Burkitt lymphoma (BL), an aggressive malignancy of germinal center B cells that significantly contributes to cancers. Increased BL risk is linked to chronic infections with *Plasmodium falciparum* infection[4] and Epstein-Barr virus (EBV)[5], and *IG*::*MYC* translocations[6,7], which are early, likely primary, cellular abnormalities[8]. Additionally, increased BL risk in SSA is associated with germline[9] and environmental risk factors (e.g., early EBV or chronic *P. falciparum* infection)[10], and somatic mutations, particularly in *MYC, TP53, ID3, TCF3, CCND3,* and *SMARCA4*[10–14]. However, the presence of other somatic chromosomal abnormalities, such as mCAs, contributing to or coinciding with *IG*::*MYC* abnormalities has not been investigated in SSA.

Recent improvements in access to genome-wide scanning of single nucleotide polymorphisms (SNPs), including in Africa[9], and refinements of computational methods to detect mCAs ≥2 Mb in size[1] facilitate research in this area. mCAs arise from the clonal expansion of cells with distinct chromosomal abnormalities, including duplications, deletions, and CNLOH[1,15–17]. mCAs in blood provide a measure of chromosomal instability in peripheral blood leukocytes[18] and their characterization has yielded insights into etiology, such as an increase in prevalence with older age and association of mCAs with an elevated risk of chronic diseases, including leukemia and select solid tumors[1,3,15,19–23].

In this work, we report a baseline prevalence of mCAs in peripheral blood samples of 8.4% in 931 children with BL, in 3.4% of 3,645 cancer-free children from Uganda, Tanzania, and Kenya[24], and in 4.5% of 177 hospital-enrolled children in Malawi with other solid (non-BL) cancers[25]. We also show in sensitivity analyses in a combined analysis of 6914 individuals from our SSA studies and US European-ancestry cancer-free adults using similar genotyping and analysis platforms that mCA prevalence in cancer-free children in sub-Saharan Africa is 4.6% compared to 5.1% in cancer-free European-ancestry adults aged 55–59

years in the US Prostate Lung Colon and Ovarian study[19] and 8.6% in cancer-free Ghanaian men aged 50-54 years.

## Results

Table 1 shows the characteristics of 931 BL cases and 3822 BL-free children enrolled in Uganda, Tanzania, Kenya, and Malawi (Fig. 1A, B). Three-quarters of the BL cases were from Uganda, Kenya, and Tanzania in the EMBLEM study, while the other 25% were from Malawi. The age of BL cases was 7.37 [SD 3.55] years and similar to that of BL-free controls (7.44 [SD 3.61] years). Males were more frequent among BL cases than BL-free controls (62% vs. 52%)[7,24]. Based on thick film microscopy or *P. falciparum* PCR, asymptomatic *P. falciparum* infection was detected in 319 (34%) BL cases and 1836 (48%) controls.

### Autosomal mCAs in peripheral blood of children
Overall, we detected 188 mCAs in 131 (3.4%) of 3822 BL-free controls compared to 250 mCAs in 78 (8.4%) of 931 BL cases. mCAs were detected at a higher average cell fraction (CF) in BL cases than in BL-free controls (0.17 [SD 0.27] vs. 0.06 [SD 0.16]). The range of CFs in BL cases overlapped with those in BL-free children (0.005–0.974 vs. 0.008–0.992; Fig. S1). Among the BL cases with detected mCAs, the mean number of mCAs per child was 1.44 [SD 2.31] and it was 3.21 [SD 3.24] in the BL-free controls, ($t$-test $P$-value = $4.50 \times 10^{-5}$, Table 2). mCAs in peripheral blood were more likely to be detected in BL cases than in BL-free controls (logistic regression test OR = 2.80, 95% CI 2.06–3.81, $P$-value = $5.70 \times 10^{-11}$, Table 2), after controlling for age, sex, country, and asymptomatic *P. falciparum* infection. Cross-sectional positivity for *P. falciparum* infection was not associated with detection of mCAs among BL-free children or BL cases.

In stratified analyses restricted to EMBLEM participants (i.e., from Uganda, Tanzania and Kenya), we detected 165 mCAs in 123 of 3645 (3.4%) cancer-free children compared to 223 mCAs detected in 70 (10%) of 701 BL cases (Table 2 and Fig. 2). The mean number of mCAs was 1.34 [SD 2.19] in controls and 3.19 [SD 2.93] in BL cases ($t$-test $P$-value = $1.15 \times 10^{-5}$, Table 2). In EMBLEM alone, logistic regression analysis showed that mCAs were more common in BL cases than in cancer-free children (OR = 3.12, 95% CI 2.27–4.30, $P$-value = $2.79 \times 10^{-12}$, Table 2). A Mantel-Haenszel stratified analysis did not show any independent effect of study countries (which also correlate with ancestry patterns) on mCA detection (OR for the association of mCAs with BL was 3.14 (95% CI = 2.31–4.27), $P$-value = $5.84 \times 10^{-14}$, Table 2). The country specific ORs for mCAs among BL cases were similar (Fisher's exact ORs 2.95–3.32, Woolf test for homogeneity, $P$ = 0.94, Table 2).

In Malawi participants (all enrolled at a tertiary-care hospital), 23 mCAs were detected in 8 (4.5%) of 177 children with non-BL cancers and 27 mCAs were detected in 8 (3.5%) of 230 children with BL (Fisher's exact OR = 0.76, 95% CI = 0.24–2.38, $P$-value = 0.62, Table 2). The average number of mCAs per child was 2.88 [SD 3.56] among those with non-BL cancers and 3.38 [SD 5.55] in those with BL ($t$-test $P$-value = 0.83, Table 2).

### Detection of mCAs on sex chromosomes
The analysis of chromosome X mCAs was performed only in girls. We detected 16 mCAs in 13 (3.7%) of 353 girls with BL compared with 44 mCAs detected in 32 (1.7%) of 1832 BL-free girls (OR = 2.15, 95% CI 1.02–4.26, Fisher's exact $P$-value = $3.74 \times 10^{-2}$, Table 3). Chromosome X mCAs were detected at a lower CF range in girls with BL (0.03 [SD 0.02], range: 0.011–0.099) than BL-free girls (0.06 [SD 0.12], range: 0.010–0.693). As observed with autosomes, in EMBLEM, chromosome X mCAs were more likely to be detected in girls with BL compared to BL-free girls (Fisher's exact OR = 3.12, 95% CI = 1.47–6.29, Fisher's exact $P$-value = $1.65 \times 10^{-3}$, Table 3). In Malawi, chromosome X mCAs were not detected in the BL cases but were seen in 3 (3.6%) children with non-BL cancers (Table 3). We did not analyze chromosome X mCAs in males because males have one copy of the X chromosome, thus, it is

## Table 1 | Characteristics of study participants in the EMBLEM and Malawi studies

| Characteristics | Burkitt lymphoma (BL) cases | Controls | Total |
|---|---|---|---|
| Country | | | |
| Uganda | 349 (37%) | 2007 (53%) | 2356 (50%) |
| Tanzania | 107 (11%) | 755 (20%) | 862 (18%) |
| Kenya | 245 (26%) | 883 (23%) | 1128 (24%) |
| Malawi[a] | 230 (25%) | 177 (5%) | 407 (8%) |
| Sex | | | |
| Male | 578 (62%) | 1990 (52%) | 2568 (54%) |
| Female | 353 (38%) | 1832 (48%) | 2185 (46%) |
| Age, years | | | |
| 0–5 | 326 (35%) | 1250 (33%) | 1576 (33%) |
| 6–10 | 413 (44%) | 1686 (44%) | 2099 (44%) |
| 11+ | 187 (20%) | 877 (23%) | 1064 (22%) |
| Missing | 5 (1%) | 9 (0.2%) | 14 (0.3%) |
| Mean age | 7.37 (SD:3.55) | 7.44 (SD: 3.61) | 7.43 (SD: 3.59) |
| *P falciparum* infection± | | | |
| Negative | 592 (64%) | 1935 (51%) | 2527 (53%) |
| Positive | 319 (34%) | 1836 (48%) | 2155 (45%) |
| Missing | 20 (2%) | 51 (1%) | 71 (1%) |

[a]Controls in Malawi were BL-free children with other non-leukemic/lymphoid solid organ cancers, whereas those in EMBLEM were cancer-free control children enrolled from the general population.
±Infection status was based on detection of parasitemia by thick film, antigenemia by rapid diagnostic test, or sensitive PCR.

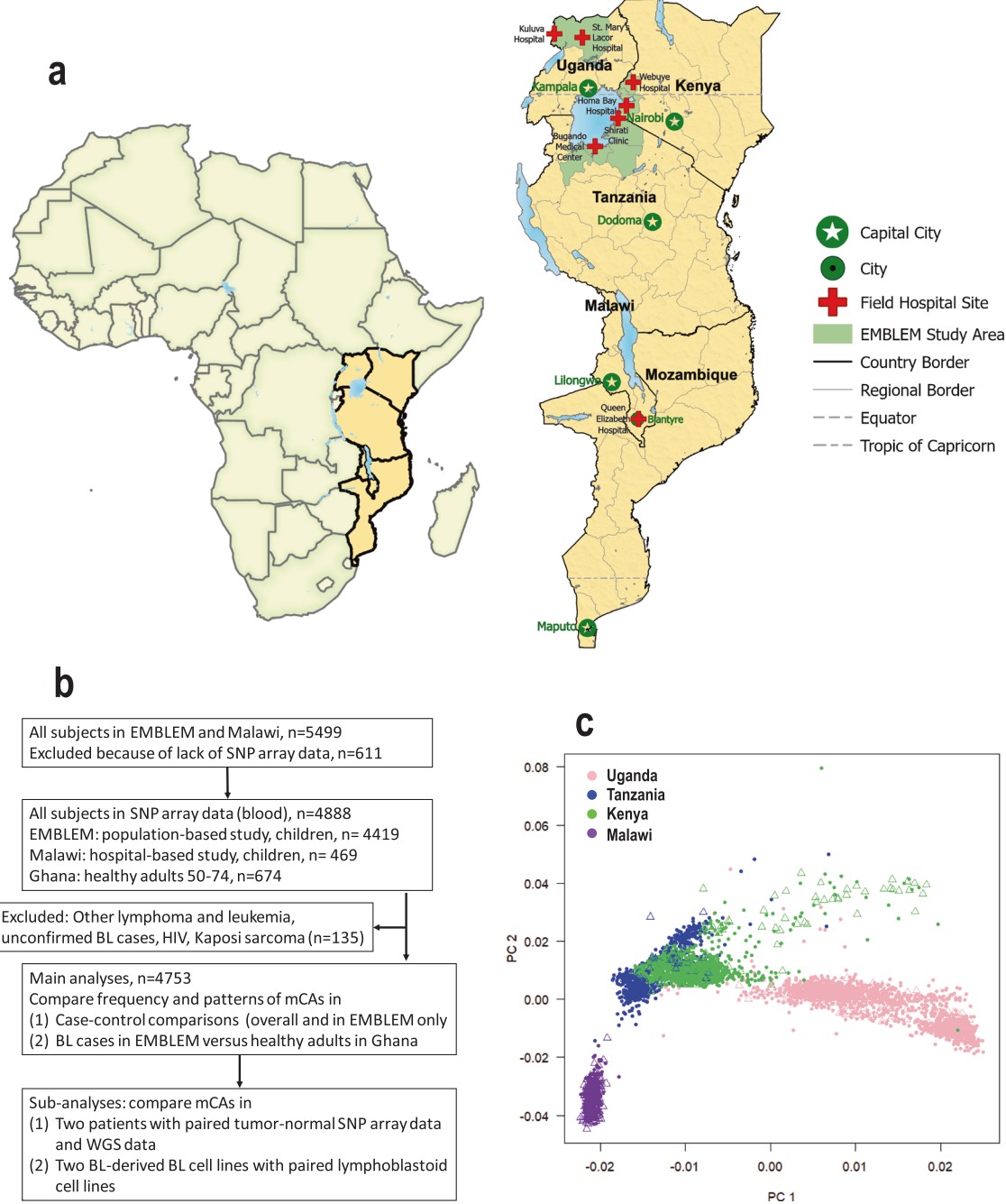

**Fig. 1 | Study participant regions and sample characteristics. a** Map showing study countries and enrollment sites. The map was drawn using ESRI ArcGIS Pro software. No portions of this figure were imported as image components from a database; **b** Flow diagram of subjects and samples analyzed; **c** population substructure in the study population based on analysis of principal components (PC) using 787,731 genotyped uncorrelated ($r2 < 0.3$) SNPs outside the HLA region (triangles represent BL cases, solid circles represent controls; 'Source data are provided as a Source Data file').

not possible to calculate the BAF values needed for mCAs detection. However, our previous work indicates that in peripheral leukocytes of males, mosaicism in chromosome X loss is rarely tolerated[26], reducing the chances of its detection.

The frequencies of Y chromosome mCAs (all loss events detected in CFs ranging from 0.007 to 0.333) were similar between BL cases and BL-free children (90 [4.5%] mCAs detected among 1990 cancer-free boys and 27 [4.7%] of 578 boys with BL; Table 3).

**mCAs in paired tumor-normal samples**

Because the presence of circulating tumor cells can result in false-positive inferences about mCAs, we carefully analyzed two of 13

informative EMBLEM patients with mCAs detected by both SNP array and WGS data in paired tumor-normal samples from the Burkitt Lymphoma Genome Sequencing Project (BLGSP)[14]. These two cases serve as technical controls to assess whether mCAs detected in tumor cells and blood match. First, we confirmed that the *IG*::*MYC* rearrangements characterized in the tumors[14] were not detected in the DNA from the corresponding peripheral blood sample of these 13 patients, based on an average WGS read depth of 40x in BLGSP WGS data[14]. For the two informative patients, we performed a more extensive re-analysis with four different structural variant callers (Table S1). Because the *IG*::*MYC* rearrangements were easily identified in tumor WGS but not in peripheral blood, this reduces the concern about false-

**Table 2 | Mosaic chromosomal alterations >=2 MB detected on autosomes of BL cases and cancer-free controls in EMBLEM and non-BL cancer controls in Malawi**

| | Burkitt lymphoma (BL) cases | | | | Controls | | | | | | |
|---|---|---|---|---|---|---|---|---|---|---|---|
| | # cases | # BL cases with mCAs (Freq.) | # mCAs (mean mCAs) | # controls | # controls with mCAs (Freq) | # mCAs (mean mCAs) | OR[a] | 95% CI | P-value | P value for mean mCAs BL cases vs. controls | |
| EMBLEM | | | | | | | | | | | |
| Uganda | 349 | 30 (0.083) | 99 (3.30) | 2007 | 62 (0.031) | 96 (1.55) | 2.95 | (1.81–4.71) | $1.44 \times 10^{-5}$ | 0.01 | |
| Tanzania | 107 | 12 (0.112) | 23 (1.92) | 755 | 28 (0.037) | 28 (1.00) | 3.27 | (1.46–6.92) | $1.99 \times 10^{-3}$ | 0.09 | |
| Kenya | 245 | 28 (0.114) | 101 (3.61) | 883 | 33 (0.037) | 41 (1.24) | 3.32 | (1.89–5.80) | $1.67 \times 10^{-5}$ | $7.78 \times 10^{-4}$ | |
| Total | 701 | 70 (0.1) | 223 (3.19) | 3645 | 123 (0.034) | 165 (1.34) | 3.12 | (2.27–4.30) | $2.79 \times 10^{-12}$ | $1.15 \times 10^{-5}$ | |
| Malawi | 230 | 8 (0.035) | 27 (3.38) | 177 | 8 (0.045) | 23 (2.88) | 0.76 | (0.24–2.38) | 0.62 | 0.83 | |
| All countries | 931 | 78 (0.084) | 250 (3.21) | 3822 | 131 (0.034) | 188 (1.44) | 2.80 | (2.06–3.81) | $5.70 \times 10^{-11}$ | $4.50 \times 10^{-5}$ | |
| P falciparum infection | | | | | | | | | | | |
| Negative | 592 | 60 (0.1) | 200 (3.33) | 1935 | 60 (0.031) | 87 (1.45) | Ref. | | | 0.001 | |
| Positive | 319 | 16 (0.05) | 46 (2.88) | 1836 | 68 (0.037) | 98 (1.44) | 1.00 | (0.73–1.36) | 0.98 | 0.14 | |
| Missing | 20 | 2 (0.1) | 4 (2.00) | 51 | 3 (0.059) | 3 (1.00) | NA | | | | |

[a]Association analysis for all countries or restricted to EMBLEM data. EMBLEM results are considered more reliable because they are from a population-based sample. Mean mCAs refers to the number of mCAs per individual among individuals that have detectable mCAs. The logistic regression analysis was adjusted for age at DNA collection, country of origin, sex, and *P. falciparum* status. The odds ratios (ORs) from country-specific analyses are based on Fisher's exact test. The overall OR is based on logistic regression. In EMBLEM, ORs were not heterogenous across countries (Woolf test *p* value = 0.94). The Mantel–Haenszel OR was 3.14 (95% CI = 2.31–4.27, *P*-value = $5.84 \times 10^{-14}$). All statistical tests used were two-sided.

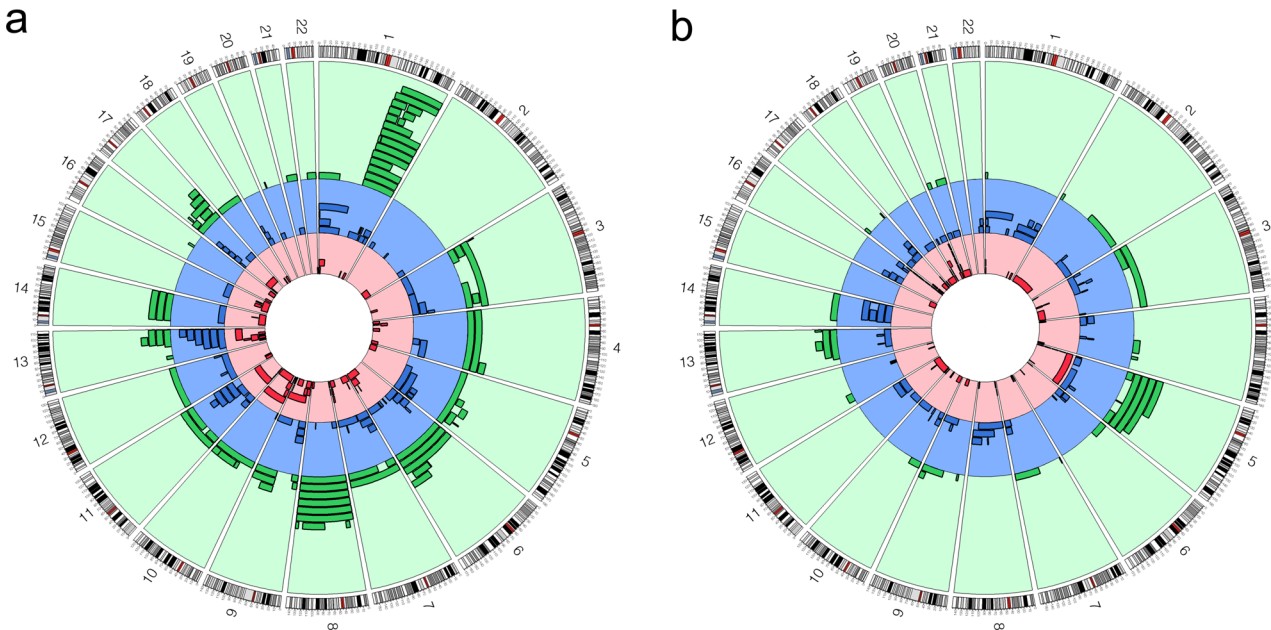

**Fig. 2 | Circos plots of detected mosaic chromosomal alterations (mCAs) and their genomic location (GRCh37).** The outer rings are for the autosomes moving clockwise from chromosome 1 to 22. The mCAs were detected in blood cells of subjects from Kenya, Tanzania, and Uganda in the EMBLEM study. The green region is mosaic duplication events; blue region is mosaic CNLOH events; red region is mosaic copy-loss events. **a** mCAs detected in 701 BL cases. **b** mCAs detected in 3645 BL-free controls; 'Source data are provided as a Source Data file'.

positive mCAs detection due to contamination of peripheral blood DNA by tumor cells. Specifically, in one patient, we detected five mCAs in the tumor WGS with CFs of ~0.82 (Fig. 3a), three of which (1q gain and losses on chromosome 3 and 6q) were also detected in the blood SNP array data with CFs ranging from 0.10 to 0.19 (Fig. 3b). The two additional mCAs detected in this patient (both CNLOHs on chr17, detected with CFs of 0.088 and 0.37, respectively) were observed only in tumor WGS (Fig. S2A) but not in the SNP array data (Fig. S2B). In this patient, more mCAs were detected in the tumor than in blood with a smaller set shared by both tumor and blood sample, possibly due to additional clonal evolution of cells that became malignant. The other BL patient had one CNLOH event on chromosome 21 detected in

normal samples (both SNP array and WGS data) at a CF of ~0.12, while this CNLOH event was not detected in tumor WGS (Fig. S3B). Thus, the clone detected in peripheral blood differed from the clone that became a tumor.

### mCAs in peripheral blood of cancer-free Ghanian men
We identified 126 unique autosomal mCAs in 96 (14.2%) of 674 cancer-free men from Ghana (4.12-fold of that observed in children) with CFs ranging from 0.006 to 0.513. The frequency of mCAs increased significantly with age, from 13% in those aged 50-54 to 26% in those aged 70–74 (logistic regression test *P*-value = 0.001). The average number of mCAs per individual was 1.31 (SD 1.04), in line with the average

**Table 3 | Mosaic chromosomal alterations >=2MB detected on female X and male Y chromosomes in BL cases and cancer-free controls in EMBLEM and non-BL cancers in Malawi**

| | Burkitt lymphoma (BL) cases | | | Controls | | | Statistic tests | | |
|---|---|---|---|---|---|---|---|---|---|
| | # cases | # BL cases with mCAs (Freq.) | # mCAs (mean mCAs) | # controls | # controls with mCAs (Freq) | # mCAs (mean mCAs) | OR[a] | 95% CI | P-value |
| Female chromosome X | | | | | | | | | |
| EMBLEM | 260 | 13 (0.05) | 16 (1.23) | 1749 | 29 (0.017) | 41 (1.41) | 3.12 | (1.47–6.29) | $1.65 \times 10^{-3}$ |
| Malawi | 93 | 0 (0.0) | 0 (0.0) | 83 | 3 (0.036) | 3 (1.00) | NA | NA | NA |
| All countries | 353 | 13 (0.037) | 16 (1.23) | 1832 | 32 (0.017) | 44 (1.38) | 2.15 | (1.02–4.26) | $3.74 \times 10^{-2}$ |
| Male chromosome Y | | | | | | | | | |
| EMBLEM | 441 | 20 (0.045) | 20 (1.00) | 1896 | 83 (0.044) | 83 (1.00) | 1.04 | (0.60–1.73) | 0.9 |
| Malawi | 137 | 7 (0.051) | 7 (1.00) | 94 | 7 (0.074) | 7 (1.00) | 0.67 | (0.19–2.33) | 0.58 |
| All countries | 578 | 27 (0.047) | 27 (1.00) | 1990 | 90 (0.045) | 90 (1.00) | 1.03 | (0.64–1.63) | 0.91 |

[a]The results are based on Fisher's exact test. All statistical tests used were two-sided.

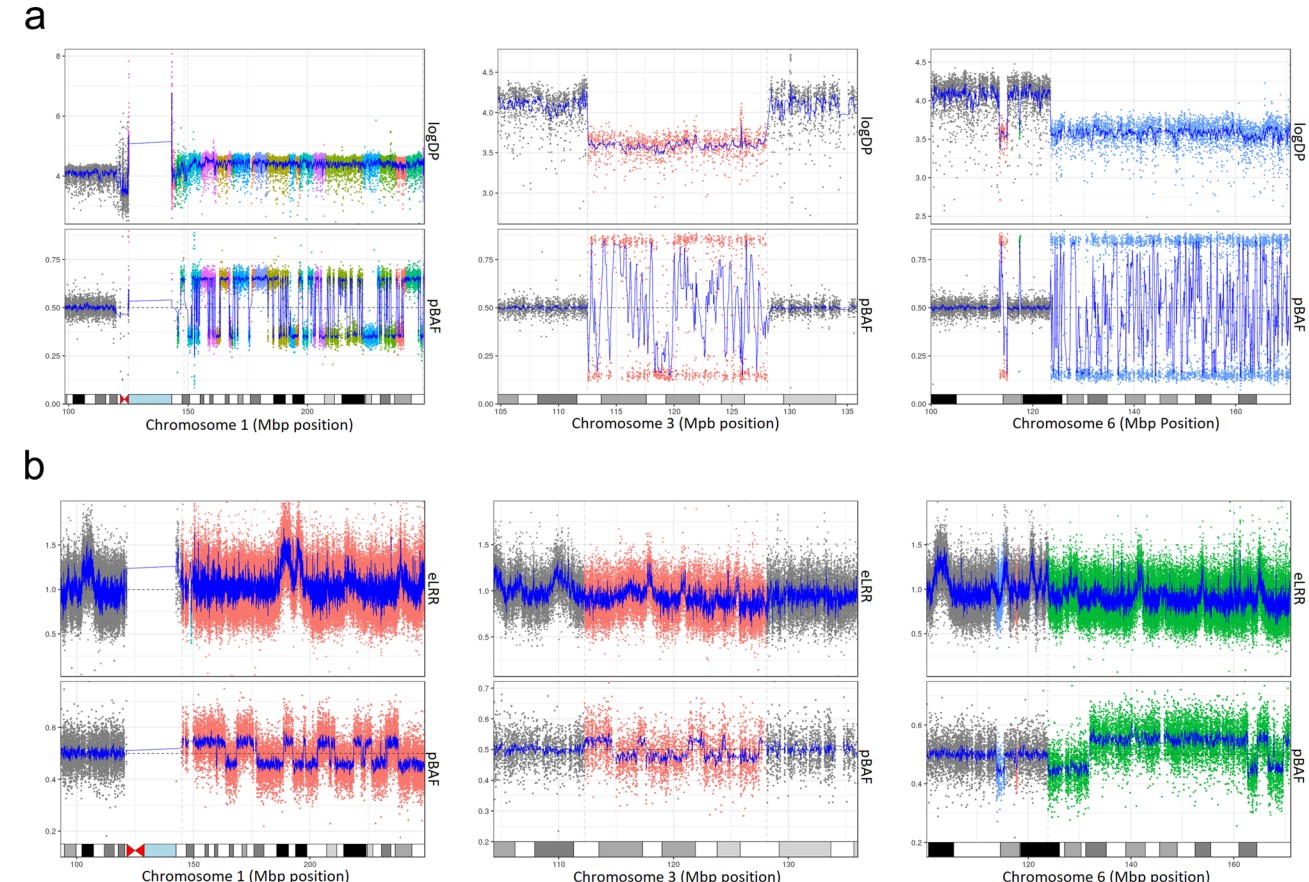

**Fig. 3 | mCAs detected in paired tumor and peripheral blood leukocytes from one of the two informative BL cases.** The figure shows three mCAs > = 2MB in size (chromosome 1q duplication, chromosome 3q deletion, and 6q deletion) detected at cell fraction (CF) ≥ 0.09 in peripheral blood leukocytes and at CF > 0.8 in tumor WGS data from the same BL patient. The mCAs were the same event types detected in similar but not identical genomic regions. **a** The 3 detected mCAs in WGS tumor data. **b** The 3 detected mCAs in SNP array data from peripheral blood leukocytes. Upper panels in the figures: Log depth of read coverage of WGS data or expo-nentiated Log R ratio (eLRR) of SNP data, respectively; Bottom panels in the figures: phase B allele frequency (pBAF) in WGS data or SNP array data, respectively. Gray color indicates regions with a normal state, all other colors (orange, blue, green) indicate regions with mCAs. 'Source data are provided as a Source Data file'.

(1.34, SD 2.19) observed in cancer-free controls in EMBLEM but lower than the average observed in BL cases.

The chromosomal distribution of mCAs in the cancer-free men differed from that observed in the BL cases (Fig. S4). The distribution of mCAs among BL cases was 37.7% of duplications on chromosomes 1 (mainly leading to gain of 1q) and 8 (Figs. 2a and S4A), 32.3% CNLOH affecting chromosomes 1, 6, and 11, and 27.4% were copy number loss events affecting chromosomes 9 and 13. One BL case had duplications

on chromosome 21. By comparison, among cancer-free children, the distribution was 37.0% CNLOH on chromosomes 10, 14, and 16, 35.2% losses affecting chromosome 19, and 18.8% gains on chromosome 5 (Fig. 2b). Among cancer-free Ghanaian men, the prevalent mCAs were duplications on chromosomes 5, 12, and 21 (Fig. S4B). The duplications on chromosome 21 spanned *RUNX1*, which we have recently identified as a susceptibility region for BL[27]. We note that in Ghanaian men, CNLOH events on chromosome 1q were prevalent while chromosome

1q gains were not seen. Among children, CNLOH events were detected on both chromosome 1p and q arms (Figs. 2 and S4A).

## Sensitivity analysis using a combined EMBLEM, Ghana, and PLCO dataset

To minimize the potential for erroneous inferences about the frequencies of mCAs in cancer-free African individuals relative to other groups that could be driven by differences in array platforms, bioinformatic filtering, and mCA detection, we performed a sensitivity analysis. We combined cases and controls from three studies ($n = 14,053$): children from EMBLEM ($n = 4888$ in total, 3645 cancer-free controls), Ghanaian men ($n = 1292$ in total, 651 cancer-free controls), and European-ancestry adults from the Prostate, Lung, Colorectal and Ovarian Cancer Screening Trial (PLCO; 7873 in total, 2618 cancer-free controls, 95.4% males). Analysis was performed on ~2.1 million shared genotyped markers using the same Illumina Infinium chemistry and analyzed in the same facility using the same pipelines. We then extracted data for cancer-free controls for the combined sensitivity analysis. Autosomal mCAs were detected in 4.6% of 3645 cancer-free controls from EMBLEM, with 4.7%, 4.5%, and 4.8% in 0–5, 6–10, and 11–16 age groups, respectively. The frequency of mCAs was 14.0% in cancer-free Ghanaian men and showed an age-related range from 8.6% in those aged 50-54 years to 17.6% in those aged 70–74 years (Fig. S5). In contrast, the mCAs frequency was 6.8% in cancer-free adults of European ancestry from PLCO, with an age-related range from 5.1% in those aged 55–59 years to 10.2% in those aged 75–79 years (Fig. S5). Overall, the frequency of mCAs in cancer-free Ghanaian men for a given age group was 2.16–2.92 times higher than in cancer-free adults of European ancestry from PLCO (Fig. S5).

## Exploratory analyses of mCA patterns in BL-derived cell lines

As access to paired tumor-normal BL and/or longitudinal samples is limited, we analyzed BL tumor-derived and corresponding non-tumor-derived lymphoblastoid cell lines (LCLs) from BL-2/IARC-304 LCL[28] and WW1-BL/LCLs[29] tumor-normal cell line pairs. These cell lines have been extensively studied to explore molecular mechanisms in BL[28,29], and could be a suitable model to gain additional insights into genome instability in peripheral blood and tumor cells of patients with BL. We authenticated the tumor-normal cell lines and confirmed genetic relatedness using microsatellite typing (Table S3). The paired BL-2/IARC-304 LCL were not informative and was not analyzed further. The paired WW1-BL/LCLs (three LCLs cultured in 2006 in Berlin (B06) and in Ulm in 2014 (U14) and in 2020 (U20) were informative and were investigated further. First, we confirmed that published t(8;14)/ IGH::MYC rearrangement[29] in the WW1-BL-derived cell lines was not detected in the WW1-LCL clones (B06, U14, and U20) tested similarly to the paired tumor-normal samples described above. Second, copy-number changes comprising gains in 3q, 8p, 14q and losses of 8q, 16p, and 20q were detected in the WW1-BL tumor-derived cell line (Fig. S6A). These gains in tumor cells were not seen in the LCLs, however, gains in 3q were identified in two of three WW1-LCL clones (-U14 and U20, but not B06, Fig. S6B). The 3q gains within the BCL6 locus differed between the tumor cell line and the -U14 and U20 LCLs (Fig. S6C), indicating that the gains in tumor and LCLs originated independently; since this region is frequently affected in BL, this might be biologically significant. Furthermore, the WGS data of the WW1 BL/ LCLs revealed uniparental disomy (UPD) on 16q in both the tumor and WW1-LCLs, where the tumor sample and LCL had lost the other parental allele other than the normal sample, further supporting the idea of independent origin of these abnormalities.

## Oncogenes and significantly mutated genes in BL overlapping mCAs on chromosome 1q, 8, and X

To gain insights into the genomic features affected by mCAs, we checked for genomic overlap between recurrent mCAs in chromosome 1q, 8, and X and known oncogenes or significantly mutated genes (SMGs) in BL. Chromosome 1q gains were the most common recurrent mCAs, with 18 gains observed in 16 BL cases (Figs. 2a and 4 and Table S4). The CFs in BL cases ranged from 0.013 to 0.199 but none of these mCAs were detected in cancer-free children (Fig. 2b). Within this region, we identified at least 9 SMGs commonly altered in BL tumors (Fig. 4)[10–14]. These include PLEKHO1, MCL1, PSMB4, ILF2, HAX1, ATP8B2, and CKS1B[30], the MIR 181B1 and MIR 181A1[31], which are overexpressed in BL. Candidate oncogenes for Burkitt lymphomagenesis, like BCL9[32], MDM4[33], members of the Fc receptor-like 1 family (FCRL), previously called immunoglobulin receptor translocation-associated (IRTA) genes[34], and IL10, which modulates EBV gene expression[35], are also located within the affected regions. Additionally, these gains on 1q are located within an oncogene-rich region, with 39 listed in the oncogene database (https://ongene. bioinfo-minzhao.org/index.html), including MCL1, CKS1B, and YY1AP1 that are significantly mutated in BL (Fig. 4)[36].

Nine chromosome 8 gains were observed in nine BL cases, making them the next most frequently observed mCA in BL cases, with CFs for the gains ranging from 0.03 to 0.67. These mCAs overlapped the BL-relevant MYC region[6,7] in seven cases and the PRKDC locus in six cases.

Inactivation of the DDX3X on the X chromosome is one of the common somatic abnormalities in BL[10,37,38]. Of 60 mCAs resulting in X chromosome loss (16 in 13 BL cases, 41 in 29 cancer-free controls in EMBLEM, and 3 mCAs from 3 children with non-BL cancers from Malawi), 37 (62%) affected DDX3X (Table S3). The distribution of the 30 mCA deletions affecting DDX3X was 11 of 13 BL cases, 16 of 29 cancer-free children, and 3 in all children with non-BL cancers in Malawi. Thus, mCA deletions resulted in loss of DDX3X in 85% of BL cases and in 55% of BL-free children, suggesting a possibility of elevated risk for BL with DDX3X loss (Fisher's exact OR 4.32, 95% CI 0.74-47.1, Fishers exact P-value = 0.09). Similar results were observed for four other genes on chromosome X, KDM6A, which is somatically involved in BL[39], and BTK, SH2D1A[40], and XIAP[41] (Table S4), which are indirectly linked with BL risk via X-linked immunodeficiencies related to EBV.

## Discussion

We detected mCAs in peripheral blood leukocytes in at least 3.4% of 3,822 BL-free children in SSA. A further elevation of mCAs frequencies to 8.4% (by 2.44-fold) was observed in 931 pediatric BL cases, and to 14.2% (4.12-fold) in cancer-free men in Ghana. Thus, our findings suggest that in SSA, mCAs could be detected in 3.4–14.2% of cancer-free individuals, with an age-related increase in adults, further elevated in children with cancer.

Our findings were confirmed in sensitivity analysis of a larger dataset of combined samples from SSA and the US that were tested in the same genotyping facility using similar array platforms and bioinformatic processing. The sensitivity analysis indicated that the relative frequency of mCAs in cancer-free children from SSA was comparable to that detected in cancer-free adults of European ancestry from the US aged 55-59 years, while the frequency of mCAs in cancer-free men from SSA was at least 2-fold higher than in cancer-free European-American adults (predominantly men) of the same age groups. These findings point to a possible role of local factors in elevating the frequency of mCAs.

The mCAs in cancer-free children from SSA commonly comprised CNLOH events on chromosomes 1, 10, 14, and 16, could be related to chronic infections prevalent in SSA. One example may be EBV, a known carcinogen for BL[38] that infects children during infancy in SSA[42] and has been shown to nonrandomly induce mutations in immortalized LCLs[43]. Another example may be P. falciparum, which stimulates B-cell proliferation mediated by cytokines, such as IL-6[44]. P. falciparum infection is recurrent, typically occurring 100-400 times per year in an average child in the study area[45], which could promote survival, expansion and diversification of cellular clones with mCAs[46]. However,

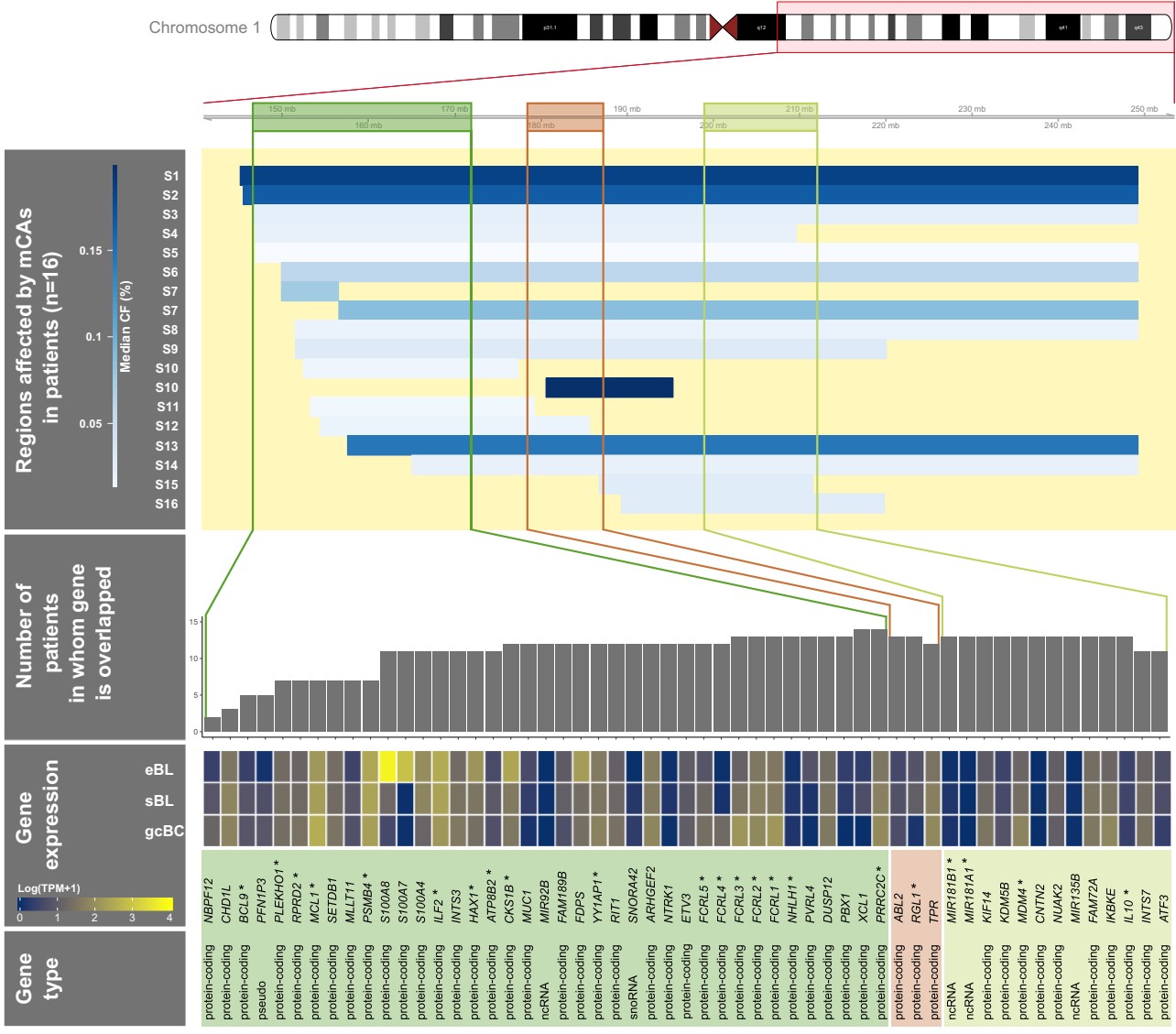

**Fig. 4 | Genomic location of mCAs gains detected on chromosome 1q in 16 BL cases.** Horizontal bar plots show regions affected by mCAs for each BL patient. Bar colors indicate median cell fraction (CF) values using a heatmap color code indicator in the bar legend. Colored boxes indicate the location of genomic regions with oncogenes from the Oncogene database and significantly mutated genes (SMGs) in BL curated from literature;[10,37] the oncogenes or SMGs in BL (marked with an asterisk) are listed below. For each gene, vertical bar plots show the number of BL cases with mCAs overlapping the oncogene or SMG. A heatmap is included to show expression of the gene or SMG in endemic Burkitt lymphoma (eBL)[68], sporadic Burkitt lymphoma (sBL)[10] and germinal center B cell populations (gcBC) for relative comparisons (see online methods for details; 'Source data are provided as a Source Data file').

we note that the chromosomal distribution of mCAs in cancer-free children differed from those seen in BL cases and included duplications on chromosome 5 (Fig. 2), which were rare in BL cases. Thus, if EBV and *P. falciparum* contribute to generating predisposing oncopathogenic mCAs, then it is logical to assume existence of some other factors that help eliminate mCAs generated by these infections, therefore protecting individuals from progressing to BL. The mCAs in cancer-free Ghanaian men were also different from those seen in cancer-free children, suggesting prevalent mCAs in cancer-free adults represent a unique biological process. For example, while mCAs involving CNLOH of 1q were frequent in cancer-free Ghanian men, 1q gains were rare, although these were frequently seen in BL cases in regions with multiple oncogenes or SMGs in BL. It is possible that accumulation of CNLOH of 1q arises due to trisomy rescue (mitotic loss of one copy of the duplicated chromosome), which counters the

evolutionary accumulation of cells with 1q mCAs gain that would otherwise increase because of increased dosage of oncogenes present in 1q.

Our findings that mCAs frequencies are elevated in children with BL, as well as those with non-BL cancers (from Malawi), suggest that the observed associations may not be BL-specific. The mCAs in BL patients were typically duplications on chromosomes 1q, 8, and in chromosome 3 in regions replete with oncogenes of SMGs in BL[37,38]. It is possible that diversification and expansion of mCA clones occur before BL development. The mCAs in BL cases affected chromosomal regions with genes that are active in the germinal center dark zone, such as *BCL6* or *MYC*[8], and those known to be significantly mutated in BL (*PLEKHO1, MCL1, PSMB4, ILF2, HAX1, ATP8B2* and *CKS1B*), which a mechanistically relevant to general or cellular predisposition to BL[47] by affecting *MYC* overexpression and DNA repair failures[48,49]. In addition,

mCAs involving loss of X chromosome affected *DDX3X* region, which is frequently altered in BL[37], in 85% of BL cases compared to 55% of cancer-free children with mCAs on the X chromosome (-1.7%). Prior evidence[21] indicates that mosaic events on the X chromosome primarily impact the inactivated X chromosome and *DDX3X* is known to escape X inactivation[50]. Thus, we infer that mCAs resulting in loss of *DDX3X* could act similarly as inactivating mutations and therefore impact BL risk in affected individuals. Although the Y chromosome carries *DDX3Y*, a paralog of *DDX3X* that may compensate for the loss of *DDX3X*[51], we found that mCA events on the Y chromosome were similar in children with vs. without BL. Thus, for BL, it is possible that mCAs affecting loci that are somatically involved in BL may be markers of predisposition to BL in an individual with mCAs or a specific cell bearing the mCA.

Our findings might be explained by three scenarios, namely that mCAs detected in peripheral blood: are contaminants by tumor cells; or indicators of cancer predisposition (BL or other cancer); or directly causal of malignant transformation. Our extensive analysis of the known *IG*::*MYC* rearrangement in primary or cultured tumor-normal pairs argues against the concern for tumor contamination of peripheral blood among the BL cases. Our findings using primary tumor-normal pairs from EMBLEM and from cultured WW1-BL-LCLs agreed in detecting a subset of mCAs in both normal and paired tumor samples, which is compatible with mCAs being detected in early BL clones. Experimental studies have shown that B cells in vitro immortalized by EBV develop non-random somatic mutations[43] induced by adenosine-induced cytosine deaminase[52]. These mutations include genomic aberrations in BL-relevant *BCL6* on chromosome 3q, further stimulating this process in affected individuals and/or cells[52]. Because our study utilizes cross-sectional data based on genotyping bulk samples, we are unable to distinguish the two hypotheses, namely, whether BL developed in a cell with a pre-existing mCA or whether mCAs represent general predisposition, i.e., BL may occur in calls lacking mCAs.

Our study has some limitations. First, our sample size of ~5000 individuals from SSA is small relative to sample sizes used in genomic studies conducted in high-income countries[53]. However, our study should be considered as a pilot to provide results about the frequency of mCAs in peripheral blood of children and adults from individuals in SSA. Larger studies are needed to replicate our initial findings, clarify associations, and improve their generalizability. Because large studies typically involve multiple populations from separate countries, our study also provides relevant data. While detailed analysis revealed population stratification in the genetic ancestry of the participants, those differences did not influence the distribution of mCAs. Second, the use of dissimilar array platforms and different bioinformatic filtering strategies in different studies complicates comparisons of results in the literature. We conducted sensitivity analyses in a combined dataset of African and US genome scan data tested in the same laboratory using similar technical methods, which permits us to make limited but valid comparisons of the relative frequency of mCAs in SSA vs. in the US. Third, we report results only for large structural chromosomal alterations ≥2MB[1,20], therefore no inferences can be made regarding mutations or smaller structural variations or the frequency of clonal hematopoiesis of indeterminate potential (i.e., CHIP)[22,23]. Fourth, our cross-sectional data limit inferences about causal associations as well as with chronic infection whose burden cannot be measured using a single time measurement, e.g., *P. falciparum*, which is recurrent up to 400 times per year in an average child in the study area[45]. Biomarkers that capture both intensity of exposure and multiplicity of strains exposed to[54] are needed to reliably evaluate the role of malaria in mCA elevation. Similarly, EBV infection, a known carcinogen for BL[38] and also known to induce oncogenic changes in B cells[52], as well as other infections prevalent in the region were not assessed in our study. These factors represent promising new directions of research

using high-throughput multiple pathogen panels, such as VirScan[55], to assess their role in mCA in SSA.

In conclusion, we report a 3.4% frequency of mCAs in BL-free SSA children, 8.4% (2.4-fold increase) in mCAs frequency in children with BL or non-BL cancers, and 14.2% in cancer-free men in Ghana. The findings in the children are comparable to those detected in cancer-free adults of European ancestry in the US, while those in cancer-free men from SSA are at least 2-fold higher than in cancer-free European-American adults. These findings point to a possible role of environmental factors in elevating the frequency of mCAs in SSA individuals. The mCAs in children with BL were gains in chromosomes 1q and 8, particularly in regions that overlap with several oncogenes and SMGs in BL. Our mCA detection approach is scalable to large cohorts with SNP genotype array data and can be utilized to further investigate the health effects associated with mCAs in SSA.

## Methods

We confirm that our research complies with all relevant ethical regulations. Specifically, ethical approval for EMBLEM was granted by Uganda Virus Research Institute (GC/127), Uganda National Council for Science and Technology (HS-816), Tanzania National Institute for Medical Research (NIMR/HQ/R.8c/Vol. IX/1023), Moi University/Moi Teaching and Referral Hospital (000536), and National Cancer Institute (10-C-N133) ethics committees. Ethical approval for the original Malawi Infections and Childhood Cancer study was granted by the Malawi College of Medicine (P.03/04/277 R) and Oxford University. Because the original Malawi Infections and Childhood Cancer study did not request participants to consent to genetic testing, special ethical approval to conduct genetic testing was obtained from the National Health Sciences Research Committee (2405). The Noguchi Memorial Institute for Medical Research Institutional Review Board (001/01-02) and the NCI (02CN240) approved the Ghana Prostate Health Survey. Written informed consent was obtained from guardians of all child participants in EMBLEM and Malawi studies, and from adult participants in the Ghana Prostate Health study. Written marked assent was obtained from children ≥7 years old in the EMBLEM study only. The tumor-normal paired cell lines used in this study were obtained from the International Cancer Genome Consortium Molecular Mechanisms in Malignant Lymphoma by Sequencing (ICGC MMML-Seq) Consortium with permission obtained from the Institutional Review Board of the Medical Faculty of the University of Kiel (A150/10) and of Ulm (349/11).

### Study population

We studied children enrolled in the Epidemiology of Burkitt Lymphoma in East African Children and Minors (EMBLEM) study (2010-2016)[24,56] and in the Infections and Childhood Cancer case-control study in Malawi (2005-2010)[25] (Fig. 1a). The EMBLEM study enrolled population-based cases and cancer-free controls aged below 16 years from six defined geographical areas (or communities) in Uganda, Tanzania, and Kenya[24]. Controls were enrolled from random villages (100 in Uganda, 100 in Kenya, and 95 in Tanzania) selected from their communities. In Malawi, children with cancers (both BL and non-BL) were enrolled at Queen Elizabeth Hospital, a tertiary-level care facility, in Blantyre in the southern region of Malawi[25]. We designated children diagnosed with BL as cases and those with non-BL cancers as controls in specific analyses, as indicated (Fig. 1b). In Malawi, participants with concurrent HIV, Kaposi sarcoma, leukemia, and other lymphoid cancers were excluded, based on the original study[25]. The participants in EMBLEM and Malawi were not compensated. However, they were provided with insecticide-treated mosquito bed nets, a complete blood count, thick blood smear to detect malaria parasites, and a stool examination to detect stool parasites.

Venous blood collected (pre-treatment) in EDTA was used to test for *P. falciparum* and for DNA extraction. In EMBLEM, *P. falciparum*

infection was assessed in the field by thick blood film microscopy/antigen-capture rapid diagnostic tests[56]. In Malawi, *P. falciparum* infection was assessed using PCR as previously reported[25]. Because EBV infection generally occurs during infancy in East Africa[57,58], all children were deemed to be EBV-infected.

To gain insights about mCAs in cancer-free adults, we analyzed genotype data from 674 cancer-free men (50-74 years) enrolled in a prostate health study in Ghana (Fig. 1b), an area endemic for BL in SSA[59].

To facilitate comparisons of our mCA data in SSA with results from high-income countries, we conducted a sensitivity analysis using a dataset of shared 2.1 million markers from cancer-free controls in EMBLEM ($n = 3645$) and cancer-free Ghanaian men ($n = 651$) vs. cancer-free adult individuals from the Prostate, Lung, Colorectal, Ovarian Cancer Screening Trial (PLCO) study ($n = 2681$, 95.4% being males)[19] that were genotyped in the same genotyping facility and bioinformatically processed using the same pipeline.

### DNA extraction and genome-wide SNP array genotyping

DNA was extracted from buffy coats obtained from the participants using the Qiagen QIAsymphony automated instrument at the Cancer Genomics Research (CGR) Laboratory, National Cancer Institute (NCI)[60]. Possible contamination of samples during sample collection through DNA extraction or DNA staging activities was evaluated prior to genotyping by an STR fingerprinting (Identifiler™) assay. Samples that were deemed contaminated from sample collection through DNA extraction or DNA staging activities were excluded per standard protocol at CGR. Samples from EMBLEM and Malawi studies were genotyped using the Illumina Infinium Omni5Exome BeadChip arrays (using 4v1-1_a, 4v1-2_A, 4v1_B and 4v1-3_A1 chips, Illumina, San Diego, CA, USA); Samples from Ghanaian men were genotyped on Illumina Infinium HumanOmni5-4v1_B and HumanOmni5Exome-4-v1-1-B; Samples in PLCO study were genotyped on Illumina Infinium HumanOmni2.5-4v1_B and HumanOmni2.5-8v1_A, following standard Illumina microarray data analysis workflow[60].

Standard quality control procedures were applied to identify and exclude samples with low sample completion rates, technical duplicates, unexpected duplicates, and low sample call rates <95%, and contamination rate >0.1, as detected by the CGR GWAS QC pipeline (Fig. S7)[60]. Possible contamination during the subsequent steps throughout the laboratory genotyping processing pipeline was assessed (Fig. S7)[60]. Sample contamination is predicted by running the tool VerifyIDintensity on each sample that passes all completion rate filters or has a median raw intensity > 6000 (internal tests show this tool to be unreliable for low-intensity data). The tool runs one sample at a time and uses a population frequency file created with 1000G population frequencies for the SNPs on the array. A threshold 0.2 contamination has been identified to reduce the ability to bin signals into the discrete clusters that are required to call separate alleles, and a threshold of >0.1 predicted contamination was selected based on benchmark studies using the 1000G reference panel to recommend sample exclusion from downstream analyses. SNPs with less than a 0.95 completion rate and excess heterozygosity ($p < 1e-6$) were excluded from modeling by both phasing and mCA detection. Genetically inferred sex was used when there was discordance ($n = 31$) with collected data forms as they were attributed to clerical errors. These filtering steps excluded 746 samples yielding an analytical data set of 4753 participants (931 BL cases and 3645 cancer-free controls and 171 other non-BL cancers) that were included in subsequent analyses (Fig. 1B). We characterized genetic ancestry of all participants by calculating principal components (PC) using 787,731 genotyped uncorrelated ($r2 < 0.3$) SNPs outside the HLA region. The PCs showed extensive population substructure among participants across the four countries (Fig. 1C). PC1 separated ancestries related to Nilotic populations in Uganda (Northern Uganda) vs. Bantu speakers (Tanzania, Kenya, Malawi) and PC2 separated ancestries related to East (Tanzania

and Kenya) vs. South Bantu speakers (Malawi). The BL cases and BL-free controls clustered together within each ancestry.

### Detection of mosaic chromosomal alterations in SNP array and WGS data

We used EAGLE2 software for phasing to infer haplotypes for SNP array data and SHAPEIT4 software for phasing to infer haplotypes in WGS data from the Burkitt Lymphoma Genome Sequencing Project (BLGSP). MoChA software (v2021-05-14) (https://github.com/freeseek/mocha) was used to detect mCAs in the SNP array and WGS data[3]. The detection method for WGS is highly similar to the SNP array data described below, with the likelihoods being replaced by beta-binomial likelihoods with $p = 1/2$ to significantly detect genotype allelic imbalances based on the number of reference and alternative read counts and intra-correlation estimated from the autosomes. MoChA uses hidden Markov models (HMM) to integrate Log R Ratio (LRR) and B Allele Frequency (BAF) as well as leverage haplotype information to detect subtle imbalances between maternal and paternal allelic fractions in cell populations. The LRR value is the normalized measure of total signal intensity; it can be interpreted as a measure of relative copy number. The B allele frequency is derived from the ratio of probe values relative to the locations of the estimated genotype-specific clusters. All detected potential events were plotted and visualized. mCA plots were reviewed (WZ) and samples that exhibited contamination patterns, including those below <0.1, such that those with low call rates and contamination were manually removed (Supplementary Table S5). We observed that all subjects detected with autosomal mCAs and females with chromosome X mCAs have a sample completion rate >0.98. Among the males with chromosome Y mCAs, all samples except two had a sample completion rate between 0.95-0.98, the rest being higher (Supplementary Table S6). Moreover, we verified that the mCA call rates were comparable between all subjects combined vs. when we restricted to those in whom mCAs were detected (Supplementary Fig. S8). We observed that using a lower threshold resulted in was more conservative estimate of mCAs rather than an inflated one (Supplementary Table S6). LRR was utilized to identify the status of specific mCAs as unbalanced duplications, deletions, and copy-neutral loss of heterozygosity (CNLOH). A deviation from the median line (denoting heterozygosity) in BAF with a relative copy number near 2 was interpreted as CNLOH; a relative copy number above 2 was interpreted as a duplication; a relative copy number below 2 is interpreted as a loss. The proportion of somatically affected cells in the sample was calculated using the method described in our previous study[1]. Due to established metrics[1], the analysis was restricted to mCAs $> = 2$ MB. We note that MoChA methods robustly detect imbalances but not the balanced reciprocal alterations typically reported in BL.

For the sensitivity analysis, we used a combined dataset from EMBLEM, Ghana, and PLCO studies with 14,053 subjects (all the BL cases and the BL-free controls in the study passed QC) to increase the sample size for improving the accuracy of phasing using the EAGLE2 software for phasing. MoChA software (v2021-05-14) (https://github.com/freeseek/mocha) was used to detect mCAs on common variants (~2.1 million) of the combined three studies. For comparison of mCA patterns in cancer-free African vs. cancer-free European-ancestry subjects, we analyzed data for 3645 cancer-free children in EMBLEM, 651 cancer-free adults in Ghana, and 2618 cancer-free adults in PLCO overall and by 5-year age groups. The Jerrfeys interval was used for the 95% confidence interval in the age vs frequency bar plot.

### Detection of *IG*::*MYC* translocation in paired tumor-normal sample WGS data from BLGSP

The Mica-SV structural variation detection pipeline (https://github.com/NCI-CGR/MoCCA-SV) with four callers (svaba, breakdancer, manta, delly) was used to confirm presence of *IG*::*MYC* translocations

in the WGS tumor data and to search for presence of tumor translocations in the WGS from the corresponding normal sample.

## Mosaic chromosomal alterations in paired BL- and lymphoblastoid-derived cell lines

The original QIMR-WW1 BL tumor-normal LCLs were created using tumor and peripheral blood samples obtained from a 5-year-old male patient with BL from Papua New Guinea[29]. The IARC-304 BL and the corresponding LCL were created using tumor and peripheral blood samples from a 7-year-old Caucasian boy[28]. The QIMR-WW1-BL/LCLs were obtained under the framework of the MMML Network[61,62] from Michael Hummel and Harald Stein (Institute for Pathology, Charite, Berlin) in 2006, while the BL-2/IARC-304 LCL were obtained from the Deutsche Sammlung von Mikroorganismen und Zellkulturen cell lines bank (courtesy of Dr. Hilmar Quentmeier). The WW1-LCL1 received in 2006, herein referred to as WW1-LCL1-B06, was expanded by creating two spontaneous clones in Ulm in 2014 and 2020, referred to as WW1-LCL2-U14 and WW1-LCL3-U20. The authenticity and genetic relationship of the tumor-LCL pairs was confirmed using short tandem repeat (STR) cell line fingerprinting (GenePrint 10 System, Promega, Madison, WI, USA) (Table S3). The t(8;14) positive *IGH*::*MYC* rearrangements and 3q alterations in WW1 BL-LCLs was studied by FISH using commercial LSI BCL6 and LSI MYC break apart, CEP6 (as control) and LSI IGH/MYC/ CEP8 probes (all Abbott) following standard procedures[61,63]. Whole genome sequencing (WGS) of the cell lines was performed in the framework of the ICGC MMML-Seq network protocols and analysis pipelines[10,64,65]. DNA methylation analysis was performed on bisulfite-converted genomic DNA using the EZ DNA Methylation kit (ZymoResearch, Irvine, CA, USA) according to the manufacturer's instructions. DNA methylation was interrogated by the Infinium HumanMethylation450 and MethylationEPIC BeadChips (Illumina Inc., San Diego, CA, USA) following the manufacturer's guidelines. The WW1-BL, WW1-LCL-U20, BL-2, and IARC-304 DNA methylation were assessed using the Infinium HumanMethylation450 BeadChip, while the WW1-LCL-U14 and LCL-B06 were run on the MethylationEPIC BeadChip. Raw IDAT files were further processed using the minfi R package[66] and normalized using 9 healthy blood samples for the 450 K and 12 for the EPIC array. Copy number profiles were generated using the conumee R package[67]. Additionally, a detailed assessment of copy number variations was performed by hybridizing DNA on a Cytoscan HD Array (Affymetrix, Santa Clara, CA, USA). The arrays were scanned and analyzed with the Chromosome Analysis Suite (ChAS) v4.3.0.71 from Affymetrix using hg19 as reference sequence.

## Genomic features of regions overlapping with mCAs on chromosome 1q

Raw RNA-seq data from 5 germinal center B-cell populations (gcBCs) and 21 solid sporadic BL (sBL) were obtained from and processed as described Lopez et al.[10]. Raw RNA-seq data from 20 endemic BL (eBL) cases were obtained from Abate et al.[68], and pre-processed using cutadapt[69] to remove adapter sequences and trim low-quality bases. Reads were aligned against hg19 using STAR[69] (v 2.6.1d, parameter: --outSAMtype BAM SortedByCoordinate −outSAMattributes Standard −outSAMstrandField intronMotif −outSAMunmapped Within −quantMode GeneCounts). Subsequently, Stringtie[70] (v 1.3.5) was used for calculation of strand specific TPMs. For visualization the R packages Gviz (version 1.40.1) and Complex Heatmap (version 2.12.1) were used.

## Statistical analysis

The association of BL with mCA status was estimated using generalized linear models to calculate odds ratios (OR) with 95% confidence intervals (CI) and Mantel-Haenszel stratified analysis to assess homogeneity of associations of BL with mCAs across the countries. Fisher's exact test was used when cell numbers were small ($n < 5$). The average number of mCAs detected per positive individual were compared between cases and controls using two-sample t-tests. Multivariable models were implemented to control for age at enrollment, country of enrollment, sex, and *P. falciparum* positivity as fixed effects. Because this is a discovery study, two-sided *P* values < 0.05 without adjustment for multiple comparisons were considered statistically significant. Genomic distribution patterns of mCAs were visualized by drawing circos plots using circos software (http://www.circos.ca/). Cell fractions (CFs), i.e., proportion of cells with mCAs vs relative copy numbers of all cells in the sample, were plotted using ggplot2 package in R.

## Reporting summary

Further information on research design is available in the Nature Portfolio Reporting Summary linked to this article.

## Data availability

The genome-wide and phenotypic data from the Burkitt Lymphoma Genome Sequencing Project, the Ghana Prostate Healthy Study, and the EMBLEM and Malawi studies are publicly available. The previously published data from the Burkitt Lymphoma Genome Sequencing Project and the Ghana Prostate Healthy Study are available under restricted access in the Genomic Data Commons; access to these studies is available through dbGaP:[14,59] BLGSP (phs000235.v20.p6) for sub-study (phs000527.v18.p6), Ghana Prostate Study (phs000838.v1.p1). The genetic data generated in this study are genotypes of participants in the EMBLEM and Malawi studies measured using Illumina Infinium Omni5Exome BeadChip arrays. The data are under controlled access (requires IRB approval and are limited to not-for-profit research). Readers can access the data by applying via dbGaP under accession link phs001705.v1.p1. The remaining data reported in the article are available within the Article, Supplementary Information, or Source Data file provided with this paper. Source data are provided with this paper.

## Code availability

No new code was generated for the purpose of the study. The code used to perform analysis is described in the relevant sections, links to access the code provided.

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

## Acknowledgements

We thank the study population and communities for their participation. We thank Ms. Janet Lawler-Heavner at Westat Inc, (Rockville, MD, USA) and Mr. Erisa Sunday at the African Field Epidemiology Network (Kampala, Uganda) for managing the study. We are grateful to the leadership of the collaborating countries and institutions for hosting local field offices and laboratories and supporting the fieldwork. We thank Ms. Laurie Buck, Dr. Carol Giffen, Mr. Greg Rydzak, and Mr. Jeremy Lyman at Information Management Services Inc. (Calverton, MD, USA) for coordinating data, and preparing data analysis files. We acknowledge the members of the tumor genetics laboratories at the Institutes of Human Genetics in Ulm and formerly Kiel (Germany) for their support, and the members of the ICGC MMML-Seq consortium for contributing to BL cell line analyses. The study was funded by the Intramural Research Program of the Division of Cancer Epidemiology and Genetics, National Cancer Institute (NCI) (Contracts HHSN261201100063C and HHSN261201100007I), the Intramural Research Program, National Institute of Allergy and Infectious Diseases (SJR), National Institutes of Health (NIH), Department of Health and Human Services, and DFG via SFB1074 (Project B9). The authors acknowledge the research contributions of the Cancer Genomics Research (CGR) Laboratory, NCI for their expertize, execution, and support of this research in the areas of project planning, wet laboratory processing of specimens, and bioinformatics analysis of generated data under NCI Contract No. 75N910D00024.

The content of this publication does not necessarily reflect the views or policies of the Department of Health and Human Services, nor does mention of trade names, commercial products, or organizations imply endorsement by the US Government. The content of this manuscript is the sole responsibility of the authors.

## Author contributions

W.Z. analyzed and interpreted SNP array and WGS data and drafted the manuscript. A.F., S.G., M.S., and M.H. generated cell line data. H.K. analyzed Burkitt lymphoma gene expression data. M.D.O., P.K., S.J.R., C.N.T., P.A.W., R.T.K., W.N.W., N.M., E.K., T.K., I.O., I.D.L., H.N., G.C., N.M., E.B., G.N.L., S.K., N.M., C.M., E.M., R.N., A.W.H., Y.T., and A.A.A. conducted field work and interpreted data. L.W.A. advised on pathology work in EMBLEM. K.B. and J.J.G. helped design the EMBLEM Study and supervised fieldwork. M.M., A.H., N.C., W.L., B.H., and K.J. performed genetic testing, bioinformatics analysis, and helped interpret data. M.H.G. performed analyses to characterize ancestry and population substructure. S.J.C. helped design EMBLEM and advised on all stages of the study. R.S., M.Y., M.J.M., and S.M.M. conceived the idea, guided data analysis, interpreted data, and edited the manuscript. L.P.O. provided critical suggestions to improve data analysis and rigorously interpret data, reviewed sensitivity analyses, and edited the manuscript. All authors commented on and approved the final draft of the manuscript.

## Funding

## Competing interests

The authors declare no competing interests.

## Additional information

[1]Division of Cancer Epidemiology & Genetics, National Cancer Institute, National Institutes of Health, US Department of Health and Human Services, Bethesda, MD, USA. [2]Cancer Genomics Research Laboratory, Frederick National Laboratory for Cancer Research, Frederick, MD, USA. [3]Institute of Human Genetics, Ulm University and Ulm University Medical Center, Ulm, Germany. [4]EMBLEM Study, St. Mary's Hospital, Lacor, Gulu, Uganda. [5]EMBLEM Study, Kuluva Hospital, Arua, Uganda. [6]Division of Intramural Research, National Institute of Allergy and Infectious Diseases, National Institutes of Health, Bethesda, MD, USA. [7]EMBLEM Study, Moi University College of Health Sciences, Eldoret, Kenya. [8]EMBLEM Study, Academic Model Providing Access To Healthcare (AMPATH), Eldoret, Kenya. [9]EMBLEM Study, Bugando Medical Center, Mwanza, Tanzania. [10]EMBLEM Study, Shirati Health, Education, and Development Foundation, Shirati, Tanzania. [11]EMBLEM Study, African Field Epidemiology Network, Kampala, Uganda. [12]Department of Pathology, The Ohio State University, Columbus, OH, USA. [13]Center for Research on Genomics & Global Health, NHGRI, National Institutes of Health, Bethesda, MD, USA. [14]Charité - Universitätsmedizin Berlin, corporate member of Freie Universität Berlin, Berlin, Germany. [15]Humboldt-Universität zu Berlin, and Berlin Institute of Health, Institute of Pathology, D-10117 Berlin, Germany. [16]Biomedical Informatics, Data Mining and Data Analytics, University of Augsburg, Augsburg, Germany. [17]Departments of Pediatrics and Surgery, College of Medicine, University of Malawi, Blantyre, Malawi. [18]Epidemiology and Cancer Statistics Group, Department of Health Sciences, University of York, York, UK. [19]Cancer Epidemiology Unit, University of Oxford, Oxford, UK. [20]Research Department, Ministry of Health, P.O. Box 30377 Lilongwe 3, Malawi. [21]Department of Genome Regulation, Max Planck Institute for Molecular Genetics, Berlin, Germany. [22]Stanford Cancer Institute, Stanford University, Stanford, Palo Alto, CA, USA. [23]Department of Pathology, University of Ghana Medical School, College of Health Sciences, P.O. Box KB 52Korle-Bu, Accra, Ghana. ✉e-mail: mbulaits@mail.nih.gov

