## [Peer Review File · Nature Communications]

REVIEWER COMMENTS

Reviewer #1 (Remarks to the Author): Expert in Burkitt lymphoma genomics

This is an interesting manuscript that identifies an unexpectedly high frequency of mosaic chromosome abnormalities in African children, and even higher in children with Burkitt lymphoma. This is an important finding. It is novel and is likely to spur further investigation into the the cause of this increased prevalence and whether/how it contributes to the pathogenesis of BL.

I have some comments:

1. The statement that the incidence in African children is higher than in western cohorts is based on a comparison with historical controls. How can the authors be sure this is not simply due to different assay or analysis pipeline?
2. The authors go to some length to show the mCA identified in children with BL is not due to contamination by tumour cells. But I am not sure they have fully addressed the converse question -ie, does BL arise from the abnormal mCA clone? There are two alternative possibilities: either the presence of the mCA represents a marker of some process that increases risk of BL (not needed to be in the same cell), or alternatively that the mCA itself predisposes cells to malignant transformation (must be in the same cell). It is not clear to me which mechanism is at play here but it is an important question that should be answerable from the data available.
3. Some more information on sex chromosome mCA would be helpful. Which regions of the chromosome are involved? Does it affect the regions subject to X-chromosome inactivation or might it lead to loss of sex chromosome genes known to be tumour suppressors. Loss of DDX3X is one of the commonest alterations in BL. Is this a potential mechanism of DDX3X inactivation?
4. Is there any evidence that mCA ends up affecting gene expression?
5. The figures are a bit underwhelming. Some of the most exciting data is hidden in the tables or text - showing more of this data graphically would make it more amenable to the reader.

Reviewer #2 (Remarks to the Author): Expert in blood and childhood cancers in Africa, and health disparities

The authors present groundbreaking findings on the potential role of chromosomal instability and etiopathogenesis of Burkitt lymphoma and other childhood cancers. The methods are clearly described

and sound; the paper is well written and balanced. There are several elements that I suggest the authors modify to improve the manuscript:

1. The title states "...of children in malaria-endemic regions of sub-Saharan Africa." Although accurate in terms of the geographical location of study subjects, emphasizing malaria endemicity in title is rather misleading because it insinuates that the observed findings are related to malaria exposure - although the authors did not test this hypothesis in this particular manuscript. There are many other potentially unique features of children that live in this part of the world and it is probably best to avoid particularly associating the findings with malaria endemicity.
2. The background section (starting on Page 3) has no section title. Please check if this is acceptable formatting for this journal.
3. In the background section, it is stated that Burkitt lymphoma is responsible for 50 - 75% of childhood cancers in sub-Saharan Africa. This is a gross over-estimation! The authors can be forgiven for stating this fallacy since it is common in older literature from Africa and remains difficult to absolutely debunk because of the absence of population-based childhood cancer epidemiology studies from Africa. Nevertheless, single and multi-center experiences with improved diagnostics over the last ten years suggest Burkitt lymphoma contributes to about 10% of childhood cancers. It may be safer for the authors to simply state that Burkitt lymphoma contributes "significantly" to childhood cancer burden rather than commit to what is becoming an obviously fallacious belief, i.e., 50 - 75%.
4. Page 3, Lines 76 - 78: I am not sure why the authors consider it important to mention that long-distance PCR and Sanger sequencing are impractical in SSA patients. However, since they bring it up, please state why it is impractical. Also, these technologies are not only used in the US and Germany - singling out these countries here is rather awkward and seems unnecessary.
5. Page 4, Line 93: "measure" the frequency of mCAs is probably a more purposeful term than "generate data."
6. I note that the authors state p-values as powers of ten, e.g., 4.50×10^{-5} . Please check that this is appropriate formatting for this journal and that there is no lower limit, e.g. $p < 0.0001$ for reporting p-values.

Reviewer #3 (Remarks to the Author): Expert in mosaic chromosomal alterations, cancer genetics and genomics

The authors of the study analyzed the presence of mCA in a total of 5,000 sub-Saharan African children, with or without Burkitt lymphoma, as well as in 674 healthy adults. Additionally, they conducted SNP array and WGS analyses on a total of 13 Burkitt lymphoma cases. They observed a higher frequency of mCA in sub-Saharan African children, and this increase was even more pronounced in Burkitt lymphoma cases. They found a significant increase in mCA in Burkitt lymphoma cases, particularly for chr1 and 8 gain, which contain many Burkitt lymphoma-related genes.

However, there are some concerns with the study. First, the authors' conclusion that the frequency of mCAs in BL-free SSA children is comparable to that observed in adults in high-income countries is problematic. The authors did not compare their results with adults in high-income countries using the same microarray, same batch, and same QC. Therefore, their main argument may be overclaimed and not supported by logical results.

Second, the number of subjects analyzed for mCA is not sufficient, and it is unclear whether mCA calls were made for all samples at once or not. The reviewer is concerned about the stability of the mCA calls, especially for controls. Generally, more samples yield more stable phasing results. In the reviewer's opinion, analyzing 4,000 or 5,000 samples is not very stable (although not too small).

Third, the QC criteria employed in this study are not strict enough. The relaxed criteria could lead to increased mCA calls.

Finally, the authors should show the distribution of mCA in adult subjects stratified by age to demonstrate the consecutive shift in mCA frequency from children to adults. This information would be valuable in understanding the age-dependent effects on mCA frequency.

Reviewer #4 (Remarks to the Author): Expert in computational cancer genomics and cancers in Africa

General Comment

The manuscript titled "Mosaic chromosomal alterations are detected at a high frequency in peripheral leukocytes of children in malaria-endemic regions of sub-Saharan Africa" by Zhou et al. provides insightful findings on the frequency and patterns of mosaic chromosomal alterations (mCAs) in paediatric Burkitt lymphoma (BL) cases and control children from sub-Saharan Africa (SSA). Overall, the study is well-conducted and provides valuable insights into the prevalence and characteristics of mCAs in SSA. While I appreciate the researcher's efforts in presenting a comprehensive report in this manuscript, several limitations and areas for improvement could be addressed. I understand that some of my suggestions may not be feasible due to the complexity, time, and cost of collecting additional data. Nonetheless, it would be worthwhile to address the concerns if the researcher can obtain data from other resources or if these data are readily available.

Comments

1. Address the issue of population stratification: The researchers could perform a more detailed analysis of population stratification in their study population, including assessing the genetic ancestry of the participants. This could help to rule out the possibility that differences in mCA frequencies between populations are due to genetic differences rather than environmental or lifestyle factors.

2. The researchers could consider expanding the study population to include participants from other regions of SSA and other parts of the world to increase the generalizability of their findings. Additionally, the researcher could investigate potential differences in mCA frequency between urban and rural populations within SSA, as differences in environmental exposures and lifestyle factors may contribute to differences in mCA frequency between these populations.

3. Replicate the findings in other populations: Replicating the findings in other populations could confirm the validity of the results and provide further insight into the factors contributing to mCA development.

4. The study would benefit from including additional control groups, such as children with other types of haematologic cancer or non-cancerous diseases, to better understand the specificity of the observed mCA patterns in SSA. This could help to determine if mCA frequencies are specific to Burkitt lymphoma or if they are more broadly associated with haematological malignancies or cancer in SSA.

5. It would be interesting to investigate the presence of mCAs in other tissues, such as bone marrow, to understand better the relationship between mCAs and haematological malignancies in the SSA population.

REVIEWER COMMENTS

Reviewer #1 (Remarks to the Author): Expert in Burkitt lymphoma genomics

This is an interesting manuscript that identifies an unexpectedly high frequency of mosaic chromosome abnormalities in African children, and even higher in children with Burkitt lymphoma. This is an important finding. It is novel and is likely to spur further investigation into the cause of this increased prevalence and whether/how it contributes to the pathogenesis of BL.

I have some comments:

1. The statement that the incidence in African children is higher than in western cohorts is based on a comparison with historical controls. How can the authors be sure this is not simply due to different assay or analysis pipeline?

Response: We agree with the reviewer's concern, which was also raised by reviewer #3. We have removed this statement as well as references #2 (PMID: 25634561) and #21 (PMID: 29995854). To enable more direct comparisons of the mCA frequency observed in Africans versus that observed in adults in high-income countries, we conducted additional, sensitivity analyses using data from three populations genotyped at the NCI Cancer Genomics Research Laboratory (CGRL). CGRL has genotyped hundreds of thousands of samples in the last 20 years and follows rigorous robust standard operating procedures for population genetic studies.

The populations/samples used in the sensitivity analysis are described in the 1st paragraph on page 19 and the 2nd paragraph on page 21 in the Online Methods and the results are reported in a new section titled "Sensitivity analysis using a combined EMBLEM, Ghana, and PLCO dataset" on page 9 and in a new Supplementary Figure S5. Briefly, we used a combined dataset of 14,053 (4,888 EMBLEM/Malawi children, 1,292 Ghanaian adults, and 7,873 from the Prostate, Lung, Colorectal and Ovarian Cancer Screening Trial (PLCO)). The data in EMBLEM, Malawi, and Ghana were genotyped on Illumina Infinium HumanOmni5 and HumanOmni5Exome; Samples in PLCO study were genotyped on Illumina Infinium HumanOmni2.5 which contains high overlap with the Infinium HumanOmni5 arrays. Phasing was performed using EAGLE2 using a common set of 2.1 million SNPs among 14,053 samples. A total of 6,914 cancer-free individuals from the three populations (3,645 children, 651 adults from Ghana, and 2,618 adults from PLCO) were scanned for mCAs using MoChA. This combined analysis allowed us to check the robustness of our findings and to also compare our results from African children and adults to those from European adults using results generated under the same technical procedures. The summary of our results is as follows:

- a. mCAs were detected in 4.6% of the cancer-free children in EMBLEM. This is slightly higher than 3.4% reported previously in our manuscript, suggesting that our initial results were conservative. We attribute this difference to better phasing due to the larger dataset. This issue was commented on by Reviewer #4, below. We report our original analysis in the manuscript as the primary findings and also report the sensitivity results to facilitate comparisons across the three populations.
- b. Using the results from the sensitivity analysis, we make the following comparisons of the results from Africa with those of European-ancestry individuals living in the US as follows.
 - i. The overall autosomal mCAs frequency was 4.6% in cancer-free children in Africa, 14% in cancer-free adult African adult from Ghana, and 6.8% in cancer-free European-ancestry individuals in PLCO. The prevalence of mCAs in cancer-free African children was comparable to the frequency observed among 55-59-year cancer-free European-ancestry adults in PLCO (4.6% versus 5.1%). Among cancer-free adult men, the frequency of mCAs was ~2-fold higher in the African men from

Ghana compared to cancer-free European-ancestry adults in PLCO, who were mostly male. In both adult African and PLCO participants, the frequency of mCAs increased with age (Figure S5).

The manuscript has been revised accordingly.

2. The authors go to some length to show the mCA identified in children with BL is not due to contamination by tumour cells. But I am not sure they have fully addressed the converse question -ie, does BL arise from the abnormal mCA clone? There are two alternative possibilities: either the presence of the mCA represents a marker of some process that increases risk of BL (not needed to be in the same cell), or alternatively that the mCA itself predisposes cells to malignant transformation (must be in the same cell). It is not clear to me which mechanism is at play here, but it is an important question that should be answerable from the data available.

Response: We appreciate Reviewer #1 raising these two hypotheses that mCAs might be a marker of general (individual) or cellular (tumor occurs in the cell with mCA) predisposition to BL. Because our study utilizes cross-sectional data based on genotyping bulk samples, we are unable to definitively test two hypotheses. Our careful analysis of genomic data from patients with paired tumor-normal samples demonstrate that a subset of mCAs is detected in both normal peripheral blood and tumor compartments, whereas others are detected in samples from one compartment but not from the sample from the other compartment suggests that mCAs occur prior to BL and their detection in cancer-free children may be a global marker of genomic instability as suggested in the literature (PMID: 34099924).

It is possible that the cell with mCA is the one more likely to progress to tumor. We are unable to directly test this hypothesis in the population data. However, findings from studies of mutational patterns in lymphoblastoid cell lines suggest that development of *BCL6* mutations, which is frequently mutated in BL and other aggressive lymphomas, modulates development of additional mutations in the affected cells, which develop a higher burden of oncogenic mutations. Specifically, Caballero and Koren, 2023 (<https://doi.org/10.1016/j.xgen.2023.100305>) reported that EBV positive B-cells with *BCL6* mutations modulated mutations during subsequent cell divisions (<https://doi.org/10.1016/j.xgen.2023.100315>), potentially influence risk of malignancy in the affected cell.

To examine this possibility in the context of our study, we analyzed cultured cell lines of paired BL tumor and lymphoblastoid cell line created from the corresponding normal sample obtained from the same BL patient from BL-2/IARC-304-LCL and WW1-BL/LCLs cell lines. These cells lines are well characterized and have been used extensively to characterize the somatic landscape of BL. The WW1-BL/LCLs (with three different clones generated in 2006 in Berlin (B06), in 2014 in Ulm (U14) and in 2020 in Ulm (U20) were informative and the findings are similar to those observed in the 13 paired primary tumor-normal EMBLEM samples tested by the Burkitt Lymphoma Genome Sequencing Project. In both, a subset of abnormalities is detected in both normal and the paired tumor sample, suggesting that these abnormalities may be detected in early BL clones. Nonetheless, our data are not able to resolve whether BL developed in a cell with a specific prior mCAs, which requires both longitudinal data and single cell sequencing data to resolve.

We have revised our manuscript to describe these data on page 10 and Figure S6, provided methodological details in the Online methods on page 21-22, and included a balanced discussion highlighting the findings as well as the limitations of data.

3. Some more information on sex chromosome mCA would be helpful. Which regions of the chromosome are involved? Does it affect the regions subject to X-chromosome inactivation or might it lead to loss of sex

chromosome genes known to be tumour suppressors. Loss of *DDX3X* is one of the commonest alterations in BL. Is this a potential mechanism of *DDX3X* inactivation?

Response: We thank the reviewer for this suggestion. We have expanded our description of the results of mCAs on sex chromosomes. We clarify that these analyses were performed separately in the boys and girls. In addition, we added details about the overlap between mCAs on chromosome X and genes either mutated in BL, like *DDX3X*, or that are indirectly associated with BL via x-linked immunodeficiency syndromes, like *KDM6A*, *BTK*, *SH2D1A*, and *XIAP*. We have also added new references about what is known about lyonization of the chromosome X. Specifically, prior evidence (Machiela Nat Commun 2016: PMID: 27291797) indicates that mosaic events on the X chromosome primarily impact the inactivated X chromosome. Thus, mCAs on the X chromosome likely lead to losses of the active gene. In the case of *DDX3X*, which is known to escape X inactivation (Tukianinen Nature 2017: PMID: 29022598), loss of the inactive X chromosome would potentially alter *DDX3X* levels, potentially impacting BL risk. Our results for the genes evaluated on chromosome X seem to support this notion. These results are now reported on pages 11-12 and provided in Table S4 and discussed in the second paragraph on page 14.

4. Is there any evidence that mCA ends up affecting gene expression?

Response: We appreciate this important question by the reviewer. We did not specifically investigate gene expression in the BLGSP. Although expression data are available in that sample set, only two patients with mCAs were also investigated in the BLGSP, making this sample inadequate for detailed examination. However, based on prior reports (Thompson Nature 2019 : PMID: 31748747), mosaic loss of the Y chromosome results in loss of expression of genes located on the Y chromosome as well as across the autosomes. We agree that this will be an interesting issue to study in the future. Nonetheless, we have included gene expression data obtained from sporadic and endemic Burkitt lymphoma tumors as well as germinal center B cell populations for the genes overlapping the mCAs in Figure 4 to describe initial patterns.

5. The figures are a bit underwhelming. Some of the most exciting data is hidden in the tables or text – showing more of this data graphically would make it more amenable to the reader.

Response: We appreciate the reviewer's advice. We have included a new Figure 4 to illustrate results previously presented in Table S2 (retained in this revision) and Figure S5A and B (which have been dropped from the revision). We have kept the circos plot as a simple but useful overview of the data.

Reviewer #2 (Remarks to the Author): Expert in blood and childhood cancers in Africa, and health disparities

The authors present groundbreaking findings on the potential role of chromosomal instability and etiopathogenesis of Burkitt lymphoma and other childhood cancers. The methods are clearly described and sound; the paper is well written and balanced. There are several elements that I suggest the authors modify to improve the manuscript:

1. The title states "...of children in malaria-endemic regions of sub-Saharan Africa." Although accurate in terms of the geographical location of study subjects, emphasizing malaria endemicity in title is rather misleading because it insinuates that the observed findings are related to malaria exposure – although the authors did not test this hypothesis in this particular manuscript. There are many other potentially unique features of children that live in this part of the world, and it is probably best to avoid particularly associating the findings with malaria endemicity.

Response: We agree with the reviewer's comment about our title and have now revised to "Mosaic chromosomal alterations are detected at a high frequency in peripheral blood leukocytes of children in sub-Saharan Africa."

2. The background section (starting on Page 3) has no section title. Please check if this is acceptable formatting for this journal.

Response: We confirmed that the formatting used is acceptable.

3. In the background section, it is stated that Burkitt lymphoma is responsible for 50 – 75% of childhood cancers in sub-Saharan Africa. This is a gross over-estimation! The authors can be forgiven for stating this fallacy since it is common in older literature from Africa and remains difficult to absolutely debunk because of the absence of population-based childhood cancer epidemiology studies from Africa. Nevertheless, single and multi-center experiences with improved diagnostics over the last ten years suggest Burkitt lymphoma contributes to about 10% of childhood cancers. It may be safer for the authors to simply state that Burkitt lymphoma contributes "significantly" to childhood cancer burden rather than commit to what is becoming an obviously fallacious belief, i.e., 50 – 75%.

Response: We agree with this observation. We have revised changed the text to "an aggressive malignancy of germinal center B cells that contributes significantly to childhood cancers" in the first paragraph on page 3.

4. Page 3, Lines 76 – 78: I am not sure why the authors consider it important to mention that long-distance PCR and Sanger sequencing are impractical in SSA patients. However, since they bring it up, please state why it is impractical. Also, these technologies are not only used in the US and Germany – singling out these countries here is rather awkward and seems unnecessary.

Response: We referred to long-range PCR and Sanger sequencing as impractical for use to investigate detection of translocations in peripheral blood of participants in SSA, as has been done in the US and Germany, because the infrastructure and costs associated with these technologies present a barrier for wide adoption in studies conducted in SSA. However, this mention seems to distract readers unnecessarily, so we have deleted this mention from the paper.

5. Page 4, Line 93: "measure" the frequency of mCAs is probably a more purposeful term than "generate data."

Response: We revised the sentence to reflect the suggested edit in the last paragraph of the introduction on page 4.

6. I note that the authors state p-values as powers of ten, e.g., 4.50×10^{-5} . Please check that this is appropriate formatting for this journal and that there is no lower limit, e.g., $p < 0.0001$ for reporting p-values.

Response: We have confirmed the scientific notation used to format p-values is appropriate formatting for the journal.

Reviewer #3 (Remarks to the Author): Expert in mosaic chromosomal alterations, cancer genetics and genomics

The authors of the study analyzed the presence of mCA in a total of 5,000 sub-Saharan African children, with or without Burkitt lymphoma, as well as in 674 healthy adults. Additionally, they conducted SNP array and WGS analyses on a total of 13 Burkitt lymphoma cases. They observed a higher frequency of mCA in sub-Saharan African children, and this increase was even more pronounced in Burkitt lymphoma cases. They found a significant increase in mCA in Burkitt lymphoma cases, particularly for chr1 and 8 gain, which contain many Burkitt lymphoma-related genes.

1. However, there are some concerns with the study. First, the authors' conclusion that the frequency of mCAs in BL-free SSA children is comparable to that observed in adults in high-income countries is problematic. The authors did not compare their results with adults in high-income countries using the same microarray, same batch, and same QC. Therefore, their main argument may be overclaimed and not supported by logical results.

Response: We agree with this concern, which was also expressed by reviewer #1. We have addressed this concern by deleting all comparisons based on historical data published in high income countries. However, to facilitate comparisons and address technical issues, we performed additional extensive analyses for three populations that were tested under the same technical conditions at NCI. The new results from these analyses permit us to compare our results from individuals in sub-Saharan Africa with individuals in the US, a high income country. We have revised the manuscript as described above in our response to Reviewer #1 (on the 1st paragraph on page 19 and the 2nd paragraph on page 21 in the Online Methods and the results are reported in a new section titled "Sensitivity analysis using a combined EMBLEM, Ghana, and PLCO dataset" on page 9 and in a new supplementary Figure S5).

2. The number of subjects analyzed for mCA is not sufficient, and it is unclear whether mCA calls were made for all samples at once or not. The reviewer is concerned about the stability of the mCA calls, especially for controls. Generally, more samples yield more stable phasing results. In the reviewer's opinion, analyzing 4,000 or 5,000 samples is not very stable (although not too small).

Response: Our sample size of nearly 5,000 participants from Africa is the largest where mCA analysis has been done in sub-Saharan African individuals. We agree that this is less than what is typically reported in studies conducted outside Africa. This issue arises from a much bigger problem related to under-representation of African participants in GWAS studies (PMID: 3090154). We view our results as providing new data that could encourage more genetic research in individuals from sub-Saharan Africa to increase their representation and precision of findings from those populations. We revised the manuscript to acknowledge this limitation in the second paragraph on pages 15-16.

We also addressed this issue in our sensitivity analysis, where phasing was performed on a larger sample of 14,053 used in our sensitivity analysis and mCA detection repeated in 6,914 cancer-free individuals from the three populations (3,645 children, 651 adults from Ghana, and 2,618 adults from PLCO). Our results from these sensitivity analyses suggest the mCA frequency is slightly higher, which we attribute to better phasing. Thus, the inferences we made using the original results are likely valid and based on estimates that are conservative, perhaps because of a slightly lower quality of phasing. Since the conclusions remain unchanged, we report the original results and provide the new results as sensitivity analyses to help readers evaluate our data.

3. The QC criteria employed in this study are not strict enough. The relaxed criteria could lead to increased mCA calls.

Response: We are not sure which specific criteria the reviewer considers relaxed. Our data were subjected to established and rigorous filtering criteria of GWAS data that is performed routinely by the Cancer Genomics Research Laboratory. This laboratory has >20 years handling population genotype data of this

type. Nonetheless, we have updated Figure S7, which summarizes the key quality control procedures used in to process the genetic data used in this study. The key steps include filtering samples that do not meet the following criteria: completion call rate ≥ 0.95 ; contamination rate $< 10\%$; unexpected, duplicated samples, whereby the duplicate with lower completion rate is filtered; exclusion of SNPs with less than a 95% completion rate and excess heterozygosity ($p < 1e-6$). We have also revised the manuscript Online Methods on page 18-19.

We agree with the reviewer that calling of mCAs must balance the concern between removing noise at the expense of removing potentially true event calls. As an added QC step to remove potential false positive calls, diagnostic plots for all resulting MoChA calls were manually reviewed by the two of the co-authors, Drs. Weiyin Zhou and Mitchell Machiela, who are experienced in calling mCAs, to visually validate the mCAs included in the downstream analyses.

4. Finally, the authors should show the distribution of mCA in adult subjects stratified by age to demonstrate the consecutive shift in mCA frequency from children to adults. This information would be valuable in understanding the age-dependent effects on mCA frequency.

Response: We thank the reviewer for this suggestion, which was also made by Reviewer #1. As part of the sensitivity analyses, we plotted the frequency of mCAs by 5-year age groups and include this as Supplementary Figure S5. The results are also reported on page 9.

Reviewer #4 (Remarks to the Author): Expert in computational cancer genomics and cancers in Africa

General Comment

The manuscript titled "Mosaic chromosomal alterations are detected at a high frequency in peripheral leukocytes of children in malaria-endemic regions of sub-Saharan Africa" by Zhou et al. provides insightful findings on the frequency and patterns of mosaic chromosomal alterations (mCAs) in paediatric Burkitt lymphoma (BL) cases and control children from sub-Saharan Africa (SSA). Overall, the study is well-conducted and provides valuable insights into the prevalence and characteristics of mCAs in SSA. While I appreciate the researcher's efforts in presenting a comprehensive report in this manuscript, several limitations and areas for improvement could be addressed. I understand that some of my suggestions may not be feasible due to the complexity, time, and cost of collecting additional data. Nonetheless, it would be worthwhile to address the concerns if the researcher can obtain data from other resources or if these data are readily available.

Comments

1. Address the issue of population stratification: The researchers could perform a more detailed analysis of population stratification in their study population, including assessing the genetic ancestry of the participants. This could help to rule out the possibility that differences in mCA frequencies between populations are due to genetic differences rather than environmental or lifestyle factors.

Response: We thank the reviewer for this suggestion. We have characterized genetic ancestry of all participants by calculating principal components (PC) using 787,731 genotyped uncorrelated ($r^2 < 0.3$) SNPs outside the HLA region. The PCs showed extensive population substructure among participants across the four countries, as shown in the attached figure that is part of another manuscript under preparation to report ancestral patterns in the Burkitt lymphoma belt populations. PC1 separated ancestries related to Nilotic populations in Uganda (Northern Uganda) vs. Bantu speakers (Tanzania, Kenya, Malawi) and PC2 separated ancestries related to East (Tanzania and Kenya) vs. South Bantu speakers (Malawi). The BL cases

and controls cluster together within ancestry. The similar ancestral patterns between the BL cases and BL-free children do not support the idea that the mCA patterns observed here (the associations of mCAs with BL were similar in Uganda, Kenya, and Tanzania, although these countries have different ancestral profiles as explained above) are due to ancestral patterns or population substructure. There are differences in the patterns of association of mCAs with BL in Malawi, which we attribute to the use of hospital-based cases with BL or non-BL cancers, both groups appearing to have similarly high mCAs. We have revised the manuscript by adding these details on page 20, and also added an ancestry plot (Figure 1C) to help readers see the ancestry pattern of the cases and controls across the study, and added a sentence in the limitations section of the manuscript stating that the mCA frequency was not affected by genetic ancestry in Africans.

2. The researchers could consider expanding the study population to include participants from other regions of SSA and other parts of the world to increase the generalizability of their findings. Additionally, the researcher could investigate potential differences in mCA frequency between urban and rural populations within SSA, as differences in environmental exposures and lifestyle factors may contribute to differences in mCA frequency between these populations.

Response: We thank the reviewer for this suggestion. This is indeed our goal for future studies to comprehensively investigate potential differences in mCA frequency between urban and rural populations within SSA to gain some insights about the effects of environmental exposures and lifestyle factors. However, due to the complexity of setting up such studies and analyzing data generated from different platforms, this goal cannot be addressed within the context of this manuscript. However, this report, if published, will serve as an example and incentive for our group or others to do the research proposed by the reviewer. We reiterate this point in the Discussion, when addressing limitations.

3. Replicate the findings in other populations: Replicating the findings in other populations could confirm the validity of the results and provide further insight into the factors contributing to mCA development.

Response: As noted in our response to the comment above, we agree with this suggestion as well. The main challenges of replicating genetic findings in Africa is finding suitable datasets with data that can be used for replication because of under-representation of Africans in GWAS studies (PMID: 3090154). Nonetheless, this is our goal, and we are looking for opportunities to replicate and expand our findings to other populations in Africa, with sharable data that is compatible with study designs and technical platforms. Fortunately, recent changes by funding agencies to encourage data sharing are making this aim easier and we suspect as ongoing studies by MalariaGen, H3Africa, and MADCap become public, there will be sufficient data available to replicate our findings.

4. The study would benefit from including additional control groups, such as children with other types of haematologic cancer or non-cancerous diseases, to better understand the specificity of the observed mCA patterns in SSA. This could help to determine if mCA frequencies are specific to Burkitt lymphoma or if they are more broadly associated with haematological malignancies or cancer in SSA.

Response: We agree with this suggestion of expanding the phenotypes to study if mCA frequencies are specific to Burkitt lymphoma or if they are more broadly associated with hematological malignancies or cancer in SSA. Because our Malawi study included BL cases and children with other cancers, our study suggests that the association with mCAs may not be specific to BL. We agree that increasing this component of other phenotypes is the next step to resolving this issue. Again, this issue cannot be addressed quickly, given the complexity of negotiations involved with sharing genetic data, but we are hopeful that publication of our paper will make this goal easier to achieve.

5. It would be interesting to investigate the presence of mCAs in other tissues, such as bone marrow, to understand better the relationship between mCAs and haematological malignancies in the SSA population.

Response: We thank the reviewer for this suggestion. While the study of mCAs in bone marrow would be difficult for healthy children, such a case might be made for children with BL and other hematological malignancies. Indeed, bone marrow samples were collected, as part of the clinical staging of BL in EMBLEM, to determine leukemic state of the disease, but bone marrow samples were not kept. We will consider this idea for designing our next field study.

REVIEWER COMMENTS

Reviewer #1 (Remarks to the Author):

Thanks to the authors for such comprehensive responses to my comments. Their responses have addressed all my concerns. This is an interesting manuscript that I enjoyed reading. It makes an important and exciting set of novel observations that will spur discussion and further research. I support publication of the manuscript as is.

Reviewer #2 (Remarks to the Author):

The authors have adequately addressed the issues that this reviewer and other pertinent issues raised by other reviewers.

Reviewer #3 (Remarks to the Author):

While the authors have made efforts to address some of my concerns, I believe that their assertions lack robust scientific validation.

Their decision to omit the comparison between sub-Saharan children and those in high-income countries is a rational step.

However, this omission resulted in lack of support for the current title "Mosaic chromosomal alterations are detected at a high frequency in peripheral blood leukocytes of children in sub-Saharan Africa", the main message in the current manuscript, as it implies a contrast between sub-Saharan African children and those from other regions (specifically high-income countries in the current study).

The study's current findings primarily indicate a heightened frequency of mCA in children with Burkitt lymphoma (BL) compared to those without BL in sub-Saharan Africa. This outcome is expected and rational, given that BL is a type of blood cancer.

Moreover, the validity and stability of the present results are not entirely compelling. In terms of quality control (QC) metrics, the acceptance threshold of 0.95 seems lenient. Many studies opt for more stringent thresholds, such as 0.98 or even 0.99. The use of such a permissive threshold casts doubt on the overall integrity of the dataset.

The concept and calculation behind "contamination rates" are not sufficiently explained. The authors must elucidate how these values were derived and clarify why they are confident that subjects with around 10% contamination rates can yield dependable mCA calling results.

Reviewer #4 (Remarks to the Author):

The authors have addressed all my comments and suggestions satisfactory.

REVIEWER COMMENTS

Reviewer #3 (Remarks to the Author):

1. While the authors have made efforts to address some of my concerns, I believe that their assertions lack robust scientific validation. Their decision to omit the comparison between sub-Saharan children and those in high-income countries is a rational step. However, this omission resulted in lack of support for the current title "Mosaic chromosomal alterations are detected at a high frequency in peripheral blood leukocytes of children in sub-Saharan Africa", the main message in the current manuscript, as it implies a contrast between sub-Saharan African children and those from other regions (specifically high-income countries in the current study).

Response: We appreciate the thoughtful insights of the reviewer and agree with them that our study did not compare our results of mCA patterns in children from SSA with those in children in more developed countries. Accordingly, we have revised the manuscript title to "Mosaic chromosomal alterations are frequently detected in peripheral blood leukocytes of children with or without cancer in sub-Saharan Africa" for clarity. We hope our results will encourage research about the prevalence, distribution, risk factors and mechanisms of mCAs in African populations and to conduct comparative studies including populations from higher income countries.

2. The study's current findings primarily indicate a heightened frequency of mCA in children with Burkitt lymphoma (BL) compared to those without BL in sub-Saharan Africa. This outcome is expected and rational, given that BL is a type of blood cancer.

Response: We agree with the reviewer's observations. Our finding of a higher frequency of mCAs in children with Burkitt lymphoma (BL) compared to those without BL in sub-Saharan Africa is novel as prior studies of BL have not examined rates of mCAs in circulating leukocytes. It is biologically plausible as the reviewer states and as we noted in our discussion on page 14. Our findings are also unique because they included joint study of both normal and paired tumor samples in some individuals, and our finding that a subset of mCAs were detected in both compartments could indicate early BL clones when detected in normal samples, while a subset of mCAs found only in normal samples in healthy children or those with cancer could indicate presence of mCAs as a biomarker of predisposition to cancer (discussed in the fourth and fifth paragraphs of the discussion on pages 13 and 15).

We disagree with the reviewer that our results are specific to BL. We observed a similarly elevated mCA frequency in children with solid malignancies in Malawi as in those with BL. These results are presented on pages 5-6 and commented upon in the first paragraph of our discussion on page 12 and in the fourth paragraph on page 14.

While the reviewer emphasizes our results in children with cancer, we would like to draw attention to our findings of mCAs in healthy children in Africa because we believe that they are of great general interest. Specifically, our joint-analysis of data from three populations (in East Africa, Ghana, and the US) indicates that children in SSA have similar frequency of mCAs to that observed in 55–59-year-old adults in the US. This result is unexpected given the differences in age. The potential implications of our results are discussed in the second and third paragraphs on pages 12 and 13 of the discussion. We hope that reporting all our results (in individuals with or without cancer) will motivate future research to elucidate patterns, causes, and health consequences of mCAs in African individuals.

3. Moreover, the validity and stability of the present results are not entirely compelling. In terms of quality control (QC) metrics, the acceptance threshold of 0.95 seems lenient. Many studies opt for more stringent thresholds, such as 0.98 or even 0.99. The use of such a permissive threshold casts doubt on the overall integrity of the dataset.

Response: We would like to reassure the reviewer that the QC metrics used in our study do not represent a deviation from the established high standard at the NCI Cancer Genomics Research (CGR) laboratory, which is a well-recognized leader in genomics research in the US and internationally. We understand and agree that it might be prudent to select higher thresholds, such as 0.98 or even 0.99, for downstream analyses, but the general threshold used in our lab is 0.95. This reduces the chances that samples with mCAs are not differentially excluded from analyses as highly clonal mCAs spanning large genomic segments could induce probe missingness. In fact, early mCA papers from our group were motivated by samples that repeatedly failed genotype completion filters due to the presence of mCAs. To address the reviewer's concern, we re-analyzed the distribution of mCA calls at lower genotyping missingness thresholds. We observed that all subjects detected with autosomal mCAs and females with chromosome X mCAs have a sample completion rate >0.98. Among the males with chromosome Y mCAs, only 2 have a sample completion rate between 0.95-0.98. As such, we demonstrate that the genotyping data threshold used in our analysis does not inflate mCAs detected. We have summarized the results in the tables and figures below and in the updated Supplementary Materials.

Table S6: The number and percentage of subjects would be excluded based on different thresholds for sample call rates.

Call Rate	Detected autosomal mCAs		Total autosomal mCAs	no mCAs		Total
	N	%		N	%	
< 97%	0	0	209	18	0.38	4753
< 98%	0	0	209	34	0.72	4753

	Detected female X mCAs		Total female X mCAs	no mCAs		Total females
	N	%		N	%	
< 97%	0	0	45	9	0.41	2185
< 98%	0	0	45	16	0.73	2185

	Detected male Y mCAs		Total male Y mCAs	no mCAs		Total males
	N	%		N	%	
< 97%	2	0.017	117	9	0.35	2568
< 98%	2	0.017	117	18	0.70	2568

Figure S8. Box plots displaying sample call rates for all subjects in the analysis versus subjects with mCAs

4. The concept and calculation behind “contamination rates” are not sufficiently explained. The authors must elucidate how these values were derived and clarify why they are confident that subjects with around 10% contamination rates can yield dependable mCA calling results.

Response: The CGR lab routinely evaluates possible contamination of samples. Contamination of samples during sample collection through DNA extraction or DNA staging activities is evaluated and identified prior to genotyping by an STR fingerprinting (Identifiler™) assay. Samples that are deemed contaminated are excluded per standard practice in our genotyping facility. Possible contamination during the subsequent steps throughout the laboratory genotyping processing pipeline is evaluated using genotyped data. Sample contamination is predicted by running the tool VerifyIDintensity on each sample that passes all completion rate filters or has a median raw intensity > 6000 (internal tests show this tool to be unreliable for low-intensity data). The tool runs one sample at a time and uses a population frequency file created with 1000G population frequencies for the SNPs on the array. A threshold of 0.2 contamination has been identified to reduce the ability to bin signals into the discrete clusters that are required to call separate alleles, and a threshold of >0.1 predicted contamination was selected to recommend sample exclusion from downstream analyses based on benchmark studies using the 1000G reference panel. All mCA plots were reviewed (WZ) and samples that exhibited contamination patterns were manually removed. We have updated the methods to clarify the process for identifying and filtering contaminated samples on pages 18, 19, and 20. We have provided a new table listing the quality metrics for the patients with mCAs as Supplementary Table S5.

Table S5: Subjects with detected mCAs and estimated contamination ranges.

mCAs	# subjects with mCAs	# mCA detected	call rate range	contamination range*
Autosomal	209	438	0.988 to 0.999	0.011 to 0.063
Female chrX	45	60	0.988 to 0.999	0.012 to 0.017
Male chrY	117	117	0.95 to 0.999	0.010 to 0.063

* From subjects with available contamination range estimates

REVIEWERS' COMMENTS

Reviewer #3 (Remarks to the Author):

The authors did the best to address my comments.